

# Impact of Biomass Burning Aerosols (BBA) on the tropical African climate in an ocean-atmosphere-aerosols coupled climate model.

Mallet Marc[1], Aurore Voldoire[1], Fabien Solmon[2], Pierre Nabat[1], Thomas Drugé[1], and Romain Roehrig[1]

[1]CNRM, Université de Toulouse, Météo-France, CNRS, Toulouse, France
[2]Laboratoire d'Aérologie, Université Toulouse III – Paul Sabatier (UPS), CNRS (UMR 5560), Toulouse, France
**Correspondence:** Marc Mallet (marc.mallet@meteo.fr)

**Abstract.** The impact of biomass burning aerosols (BBA) emitted in Central Africa on the tropical African climate is studied using the ocean-atmosphere global climate model CNRM-CM, including prognostic aerosols. The direct BBA forcing, cloud feedbacks (semi-direct effects), effects on surface solar radiation, atmospheric dynamics and precipitation are analysed for the 1990-2014 period. During the June-July-August (JJA) season, the CNRM-CM simulations reveal a BBA semi-direct effect exerted on low-level clouds with an increase in cloud fraction of ∼5-10% over a large part of the tropical ocean. The positive feedback of BBA radiative effects on low-level clouds is found to be mainly due to the sea surface temperature response (decrease of ∼-0.5 K) associated with solar heating at 700 hPa, which increases the lower tropospheric stability. Over land, results also indicates a positive effect of BBA on the low cloud fraction especially for the coastal regions of Gabon and Angola with a potentially enhanced impact in these coupled simulations that integrate the response (cooling) of the SST. In addition to the BBA radiative effect on sea surface temperature, the ocean-atmosphere coupled simulations highlight that the oceanic temperature response is noticeable (about -0.2 to -0.4 K) down to ∼80 m depth in the JJA between the African coast and 10°W. In parallel to low-level clouds, reductions of ∼5-10% are obtained for mid-level clouds over central Africa, mainly due to BBA-induced surface cooling and lower tropospheric heating inhibiting convection. In terms of cloud optical properties, the BBA radiative effects induced an increase of the optical depth by about ∼2-3 south of the equator over the ocean. The result of the BBA direct effect and feedback on tropical clouds modulates the surface solar radiation over the whole Tropical Africa. The strongest surface dimming is over central Africa (∼-30 W m$^{-2}$), leading to a large reduction of the continental surface temperature (by ∼-1 to -2 K), but the solar radiation at the oceanic surface is also affected up to the Brazilian coast. With respect to the hydrological cycle, the CNRM-CM simulations show a negative feedback on precipitation over the West African coast with a decrease of ∼-1 to -2 mm per day. This study highlights also a persistent impact of BBA radiative effects on low-level clouds (increase in cloud fraction, liquid water content and optical depth) during the September-October-November (SON) period, mainly explained by a residual cooling of sea surface temperature over most of the tropical ocean. In SON, the feedback on precipitation is mainly simulated over the Gulf of Guinea with a reduction by ∼-1 mm per day. As for JJA, the analysis clearly highlights the important role of the slow response of the ocean in SON and confirms the need to use coupled modelling platforms to study the impact of BBA on tropical African climate.



## 1 Introduction

The Southern African Savannah is one of the major sources of biomass burning aerosols (BBA) at the global scale, and this region produces about one-third of the carbon emitted by fires worldwide (Vanderwerf et al., 2010). After their emissions over the continent, the BBA plumes are transported over the southeast Atlantic (SEA) during the June to October period every year, modifying the regional energy budget of the Tropical Africa through different complex mechanisms. First, BBA can interact directly with solar radiation and modify shortwave radiative fluxes at different atmospheric levels (at the surface, within the atmospheric column, and at the top of the atmosphere, TOA). Recently, it has been shown that BBA emitted over Central Africa absorb solar radiation more efficiently than previously thought (Zuidema et al., 2018; Wu et al., 2020; Denjean et al., 2020; Chauvigne et al., 2021) . The fact that such absorbing BBA are transported over highly reflective surfaces (low-level clouds) can generate a persistent positive TOA radiative forcing (Meyer et al., 2013; De Graaf et al., 2014; Kacenelenbogen et al., 2019; Mallet et al., 2019, 2020; Solmon et al., 2021) of opposite sign to that generally attributed to anthropogenic aerosols. Currently, global climate models struggle in representing this TOA radiative forcing singularity. Indeed, Mallet et al. (2021) have shown that ~75% of CMIP6 (Coupled Model Intercomparison Project) simulations represent a cooling at TOA over SEA due to BBA, both because of an underestimation of the absorbing capacity of BBA and a misrepresentation of the low-cloud fraction. In parallel to the radiative effects simulated at TOA, BBA also significantly reduce the solar radiation reaching the surface, by several tens of W m$^{-2}$ (Sakaeda et al., 2011; Mallet et al., 2019; Solmon et al., 2021). In addition, BBA are now thought be efficient cloud condensation nuclei (CCN) that can modify cloud microphysical properties as highlighted by Lu et al. (2018) or more recently by Diamond et al. (2022) over SEA. Finally, the solar radiation absorbed by BBA has been identified as a key process in the BBA impact on low-level clouds over SEA, and several studies using different approaches (models or satellite observations) indicate that BBA tend to increase the cloud fraction (CF) and water content of low-level clouds over SEA, through the semi-direct effect (Johnson, 2004; Wilcox, 2010; Mallet et al., 2019, 2020; Solmon et al. 2021; Herbert et al., 2020).

By modifying atmospheric thermodynamic profiles, absorbing aerosols also have the capacity to modify the atmospheric dynamics, convection and precipitation at regional and global scale. Samset (2022) indicate that the spread in simulated aerosol absorption in the most recent generation of climate models (CMIP6) can be a dominating cause of uncertainty in simulated precipitation change, globally and regionally. Based on the PDRMIP (Precipitation Driver and Response Model Intercomparison Project) exercise (Myhre et al., 2017), Samset et al. (2016) indicate that absorbing Black Carbon (BC) aerosols contribute to the most substantial uncertainties among global climate models in simulating the changes in surface temperature and precipitation. More recently, Persad et al. (2023) argue that tropical regions are particularly sensitive to hydrological-cycle change due to either local or remote aerosol emissions. Uncertainties in the precipitation response are mainly related to the different parameterisations used to represent the physical, chemical and optical aerosol properties and the dynamical processes involved from BC emission to the final radiative and climate impact. The complexity of hydrological feedbacks to the radiative forcing of absorbing aerosols also stems from the fact that it depends on both the so-called "fast" and "slow" responses of the climate system. The fast response is independent of changes in sea surface temperatures (SST) and mostly depends on instantaneous



modification in atmospheric radiative heating/cooling (O'Gorman et al., 2012). The "slow" response is mediated by changes

in SST and is strongly correlated with TOA radiative forcing. On the contrary, the effects of (scattering) sulfate aerosols dominated by the "fast" response over the tropics are relatively weak (Figure 3 in Samset et al., 2016).

In contrast to the interactions between desert dust aerosols and the hydrological cycle over Tropical Africa (Solmon, 2008, 2012; Balkanski et al., 2021), few studies have addressed the impact of BBA plumes emitted over central Africa on cloud properties, atmospheric dynamics and precipitation in the tropics. Recently, Solmon et al. (2021) and Ajoku et al. (2019)

addressed this issue using regional climate modelling and reanalysis data, respectively, and showed a significant effect, in particular with enhanced drying conditions over southern West Africa. On the methodological point of view, the analysis of the various impacts of BBA on radiation, clouds, circulation and hydrological cycle needs to be addressed using ocean-atmosphere coupled modeling systems, so as to enable the ocean surface temperature to respond to the BBA surface radiative forcings. Indeed, the SST response is crucial for understanding feedbacks on the cloud cover (Sakaeda et al., 2011), marine

boundary layer dynamics (Mallet et al., 2020) and the hydrological cycle (Solmon et al., 2021). At present, most studies that focused on the interaction between BBA and the tropical African climate used atmosphere-only models in which SSTs are prescribed. A few studies used an atmospheric model coupled to a slab ocean model (Sakaeda et al., 2011; Solmon et al., 2021; Jiang et al., 2020) to investigate the BBA and climate interactions but as discussed by Solmon et al. (2021), this approach might oversimplify the SST response to smoke aerosols and cloud cover perturbation. The authors state that the use of a fully

interactive ocean model will help refine the regional SST response and the associated mechanisms and interactions. Zhao and Suzuki (2019) explored the effects of black carbon and sulfate aerosols on global and tropical precipitation and emphasize that a slab-ocean model overestimates the cross-equatorial heat transport response in the atmosphere as compared with a fully coupled approach. Finally, Lu et al. (2023) recently indicated that the proper assessment of the SST-cloud feedbacks requires fully-coupled simulations because the SST changes alter the ocean circulation, which in turn affects SST and clouds.

In this context, the aim of this study is to investigate the interactions between the BBA emitted in Central Africa, the radiation budget at the regional scale and the various possible impacts on the tropical African climate using an ocean-atmosphere-aerosols coupled global climate model. This study focuses first on the JJA season, but also examines the impact of BBA during the SON, which has been less addressed to date. These two seasons are very important as they are the most intense in terms of emissions from biomass burning in central Africa and also corresponds to the development of the monsoon in Western Africa.

This work follows numerous preliminary studies that have tested and largely evaluated the model over this specific region in terms of BBA vertical structure and transport (Mallet et al., 2019; Doherty et al, 2022), aerosol concentrations above low-level clouds (Mallet et al., 2019, 2020; Shinozuka et al., 2020; Redemann et al., 2021; Doherty et al., 2022), induced diabatic heating (Cochrane et al., 2022) and direct/SDE radiative forcing (Mallet et al., 2019, 2020). This study also takes advantage of recent improvements in the representation of optical properties, in particular for the solar absorption induced by BBA (Drugé et al.,

90  2022).

In this context, this study relies on the use of the ocean-atmosphere global climate model (Voldoire et al. 2019), in which an interactive aerosol scheme (TACTIC, Drugé et al. 2022) has been included. This model will be used to investigate in more details the different impacts of BBA on the tropical African climate. After a detailed description of the modelling methodology





(section 2), section 3.1 examines the impact of BBA on cloud macrophysical (Cloud fraction) and microphysical (liquid water content) properties as well as the changes in cloud optical properties. The overall effect of the BBA direct effect and changes on tropical clouds on the solar surface radiative budget is also discussed. In section 3.2, the interaction between the BBA radiative effects, the lower tropospheric dynamics and precipitation during the JJA season is examined in more detail. Finally, the section 3.3 analyses the possible feedback of BBA emissions on radiation, cloud properties and the hydrological cycle during the SON season.

## 2 Method

### 2.1 The CNRM-CM model

The CNRM-CM model is a global climate model developed at CNRM, belonging to a family of coupled ocean-land-atmosphere climate models that contributed to the CMIP6 intercomparison exercice (Eyring et al., 2016). For CMIP6, two configurations have been used, the CNRM-CM6-1 model (Voldoire et al., 2019) which is the physical core coupled system and the CNRM-ESM2-1 model (Seferian et al., 2019) which is the earth system version based on the former with the addition of Earth System components (interactive aerosols, chemistry, carbon cycle and vegetation), the full carbon cycle and the ocean biogeochemistry. In this study, we use an hybrid version : the CNRM-CM6-1 physical core with the addition of the TACTIC aerosol scheme used in CNRM-ESM2-1 (in an updated version published in Drugé et al. 2022. Hereafter this model will simply be named CNRM-CM.

The physical core is composed of the atmospheric global model ARPEGE-Climat V6 (Roehrig et al., 2020), the surface modeling platform SURFEX v8 including notably the Interaction Soil-Biosphere-Atmosphere (ISBA) – CNRM version of Total Runoff Integrating Pathways (ISBA-CTRIP) coupled land surface modelling system, the bulk FLake model (Decharme et al. 2019), the ocean model NEMO v3.6 (Madec et al., 2017), the sea-ice model GELATO (Voldoire et al., 2019). The ECUME (Exchange Coefficients from Unified Multi-campaigns Estimates) iterative approach (Belamari and Pirani, 2007) is used to compute the air–sea turbulent fluxes. The spatial resolution is about 140 km in the atmosphere and land components. In the ocean and sea-ice components, the nominal resolution is 1°. ARPEGE-Climat includes 91 vertical levels from the surface to 0.01 hPa in the mesosphere. This global model uses a longwave (LW) radiation scheme based on the rapid radiation transfer model (RRTM, Mlawer et al., 1997) and a shortwave (SW) radiation scheme based on the Fouquart and Morcrette radiation scheme (FMR, Fouquart and Bonnel, 1980; Morcrette et al., 2008) with six spectral bands (whose limits are, respectively, 0.185, 0.25, 0.44, 0.69, 1.19, 2.38, and 4.00 $\mu$m). A detailed description of the physical core component can be found in Voldoire et al. (2019). In this study, the ARPEGE-Climat model includes an interactive aerosol scheme described in the following paragraph.

### 2.2 The TACTIC aerosol scheme

TACTIC is the bulk-bin aerosol scheme used in the CNRM global (Michou et al., 2020) and regional (Nabat et al., 2020) climate models. TACTIC has been developed to represent the main tropospheric aerosol species and their interactions with radiation





and clouds. The present work used an updated TACTIC version (Drugé et al. 2022) compared to that of CNRM-ESM2-1
described in detail in Michou et al. (2020). Briefly, the TACTIC aerosol scheme simulates the physical evolution of different
aerosol types that are supposed to be externally mixed (particles from different sources exist as separate particles): desert dust,
sea salt, black carbon, organic matter, brown carbon, sulfate, ammonium and nitrate particles. Further recent developments
concerning the formation of sulfate particles, the aerosol wet deposition, and the aerosol–radiation coupling are used here and
described in Drugé et al. (2022).

More specifically, the TACTIC aerosol scheme includes three size bins for desert dust and sea salt; two bins (hydrophilic
and hydrophobic) particles for organic matter (OA), brown carbon (BrC) and black carbon (BC), two size bins for nitrates, and
one size bin for sulfate ($SO_4$) and ammonium. Continental biogenic secondary organic aerosol (SOA) are not formed explicitly
but are taken into account through the climatology of Dentener et al. (2006), while oceanic biogenic SOA and aromatic SOA
are not yet considered. Aerosols can be interactively emitted from the surface (desert dust and sea salt) as a function of surface
wind and surface characteristics, or the scheme can consider external emission data sets, including those for anthropogenic
and biomass burning particles (BC, OA, $SO_2$ and $NH_3$). As described in Michou et al. (2015), a coefficient of 1.5 is applied
to organic carbon emissions in order to take into account the conversion of organic carbon into organic matter. In the present
version, the sulfate formation deals explicitly with the chemical oxidation of sulfate precursors into sulfate.

The atmospheric model represents the interactions between all aerosols and radiation (direct effect), as well as between
hydrophilic particles (OA, BrC, sulfates and sea-salt) and cloud albedo (first indirect effect; see Michou et al., 2020, for
details). The second indirect aerosol effect, which corresponds to interactions between aerosols and cloud precipitation, is not
included. With regards to aerosol–radiation interactions, TACTIC considers aerosol optical properties (extinction, SSA, and
asymmetry parameter) detailed in Drugé et al. (2022) for the wavelengths of the radiation scheme. These optical properties
depend on relative humidity, except for desert dust, BC and hydrophobic organic aerosol.

## 2.3   Design of the CNRM-CM model configuration

In this study, we intend to estimate the impact of biomass burning aerosols on the ocean-atmosphere coupled system. For
this purpose, two simulations, with and without biomass burning aerosol emissions have been carried out over the 1970-2014
period. However, when including such aerosols, the global mean surface radiative budget is altered and this modifies the global
mean state and in particular the global mean surface temperature. Therefore, it is difficult to disentangle the local effects
from feedbacks due to the global mean state change. Here, so as to isolate the local feedbacks, we developed a constrained
configuration by nudging the ocean temperature and salinity globally except in the tropical Atlantic region [31°S-26°S] (see
Figure A1). Both the reference experiment and the sensitivity experiment are nudged towards the temperature and salinity
fields of the first member of the CNRM-CM6-1 CMIP6 historical simulation from 1970 to 2014 with a time scale of 6h and
at each ocean level. Both simulations are also initialised in 1970 using the same CMIP6 historical experiment state in 1970.
Therefore, both simulations share the same sea surface temperature outside the tropical Atlantic region and the tropical Atlantic
sea surface temperature is not altered by the global mean state change. In summary, the present configuration allows to focus
on solely, the ocean-atmosphere feedbacks in the tropical Atlantic, while outside this region the atmospheric model is forced by



the same SSTs in each experiment. Both nudged coupled (named CPL_ndg) simulations are run over the full period 1970-2014
but the first 20 years are considered as a spin-up phase and we only analyse the period 1990-2014.

## 3  Results

In the results presented thereafter, all the anomalies correspond to the differences between the simulations with and without
BBA and the statistical test applied is the Wilks test (Wilks, 2006, 2016).

### 3.1  BBA "radiative perturbations" simulated in the CNRM-CM model

As shown in Figure 1a, the BBA AOD anomaly for the JJA shows a maximum located over central Africa with seasonal means
reaching ∼0.7 (at 550 nm). The transport (outflow) of BBA over the SEA is also consistent and lies on average between 0 and
15°S. Beyond 30°W, the simulated BBA AOD anomaly is rather low. At the global scale, the BBA AOD anomaly is found
to be significant over the major BBA source regions such as the Amazon, North America, Indonesia and Siberia (Figure A1).
Some negative AOD anomalies appear locally over the Arabian Peninsula, which could be due to feedbacks between BBA
radiative forcing, dynamics, precipitation and wet deposition or emission of mineral dust aerosols. Figure 1b also shows that
the simulated aerosol single scattering albedo (SSA) over central Africa in JJA is about ∼0.88 (550 nm) due to the presence of
smoke aerosols. This SSA value is consistent with those obtained recently (between ∼0.80 and 0.85 at 550 nm), in particular
in the framework of the ORACLES, LASIC, AEROCLO-sA or CLARIFY programs (Zuidema et al., 2018; Wu et al., 2021;
Redemann et al., 2021; Chauvigné et al., 2021). In parallel, the SSA of the smoke plumes during the transport over the SEA
is close to ∼0.90 (at 550 nm), in agreement with the values reported by Mallet et al. (2021). Figure 1b shows a slightly higher
SSA over the Amazon, consistent with the work of Johnson et al. (2016) or, more recently, by Holanda et al. (2023).

As shown in Figure 2a, BBA plumes create a strong radiative disturbance at the surface during the JJA season and sig-
nificantly decrease the solar radiation reaching the continental and oceanic surfaces. The highest solar direct surface forcing
is simulated over areas of high emissions and large AOD (Figure 1a). Over Central Africa, the BBA direct radiative forcing
reaches seasonal-mean values of ∼-10 to -20 W m$^{-2}$. The CNRM-CM simulations highlight radiative forcings of ∼-5 to -10
W m$^{-2}$ up to 0° longitude during JJA with a radiative effect gradually decreasing up to the Brazilian coast in agreement with
the AOD anomaly (Figure 1a). Beyond ∼30°W, the BBA surface forcing is found to be low over the Atlantic ocean (∼-1
to -2 W m$^{-2}$). The regional pattern of the BBA forcing is consistent with Sakaeda et al. (2011) but some differences in the
magnitude are observed with higher direct surface forcings (between -20 to -40 W m$^{-2}$) in Sakaeda et al., (2011) due to larger
AOD anomaly simulated in this study. The BBA surface radiative forcing simulated in the CNRM-CM model is also consistent
with Allen et al. (2019) in terms of regional pattern but with slightly lower values (-20 W m$^{-2}$ over Central Africa in Allen et
al. (2019)). In parallel, Figure 2b displays the solar JJA heating rate anomaly (in K by day) due to the BBA SW absorption.
This radiative perturbation is essential for representing the BBA semi-direct effect on cloud properties (Johnson et al., 2004).
As shown in Figure 2b, the maxima of the solar heating are simulated between 0 and 20°S with JJA seasonal-mean values
up to ∼0.5 K by day. This "heating effect" is mainly confined between 850 and 600 hPa, with maxima at ∼700 hPa which





corresponds to the altitude of transport of smoke aerosols (Figure 5a). It should be noted that the solar heating rate simulated in the CNRM-CM model is lower than values proposed by Wilcox (2010) or Mallet et al. (2020) with seasonal-mean values reaching ~1 to 1.5 K by day. This difference is possibly attributed to the under-estimation of low-level clouds over southeast Atlantic in the CNRM-CM model (Brient et al., 2019), limiting the reflection of solar radiation by clouds and hence solar
absorption by BBA plumes.

### 3.2    Effects of BBA on the tropical cloud properties and radiative budget during the JJA season

#### 3.2.1    Changes caused by BBA on the tropical ocean

The impact of the BBA radiative effects on the low-level cloud fraction (LCF, in %) is shown in Figure 3a. The CNRM-CM simulations generally show an increase in the LCF during the JJA, ranging from ~5 to more than 10% over much of the tropical
Atlantic. The LCF change maxima are located over the southeast Atlantic, with a positive anomaly higher than ~10% near the Angola coast. The BBA effect on the low cloud fraction is weaker over the Gulf of Guinea and LCF is approximately increased by 5% over this area. In addition to the cloud fraction, Figures 3b and 3d also show a BBA effect on the integrated water content and optical depth of liquid clouds over a large part of the Atlantic ocean. For the water content, the results indicate that LWP can be increased by about 0.006-0.008 kg m$^{-2}$ between 5 and 15°S, with maxima greater than 0.01 kg m$^{-2}$ along the
coast of Angola (Figure 3b). However, in contrast to the cloud fraction, the simulated effect over the Gulf of Guinea is found to be small. Figure 3b also shows a decrease in water content in the coastal zones of West Africa, which will be discussed later. Finally and consistently with the water content, Figure 3d shows an increase in cloud optical depth (COD) of the order of 1 to 3 between 5 and 15°S. In addition to BBA disurface forcing, this effect on cloud optical properties will contribute to the reduction of solar radiation at the oceanic surface.
Over the ocean, the Lower Tropospheric Stability (LTS, generally defined as the difference in potential temperature between 700 and 1000 hPa, or even SST; Klein and Hartmann 1993) is known to influence low-level cloud cover (Slingo 1987). Such studies show a remarkably good correlation between seasonal mean LTS and low-level cloudiness over the major tropical/sub-tropical stratocumulus regions, with cloudiness increasing by about 0.05 per 1 K of ΔLTS (Klein and Hartmann 1993). Over the Atlantic ocean, the absorption of solar radiation in the atmospheric layer where BBA reside (with the maximum heating
occurring at ~700 hPa, see Figure 2b) combined with the decrease in solar radiation and SST (see hereafter) is shown to impact efficiently the LTS. These two processes, detailed in the following, appear to be likely the main causes explaining the changes in cloud properties over the tropical Atlantic.
       As shown in Figure 2a, the BBA radiative direct effects alone contribute to a significant reduction in the solar radiation reaching both the oceanic and continental surfaces. Over the Atlantic Ocean and between 0 and 15°S, the combination of the
BBA direct forcing and changes in cloud properties (Figures 3a, 3b and 3d) lead to an important decrease in surface solar radiation, with a detectable impact along the Brazilian coast. Figure 3c shows that the simulated solar dimming between the African coast and 5°W is of the same order of magnitude (between -20 and -30 W m$^{-2}$) as in central Africa near the BBA sources. For longitudes beyond 5°W, the decrease in surface radiation remains significant (between -5 and -10 W m$^{-2}$) and is




almost exclusively due to the BBA semi-direct effect on low-level clouds (the BBA AOD anomaly is low above 5°W, Figure
1a). This decrease in solar radiation at the ocean surface is slightly offset by an increase in LW radiation (up to +10 W m$^{-2}$)
due to the positive response of the cloud cover. However, this last effect is mainly concentrated off the coast of Angola (Figure
A2 in Appendix). The surface dimming simulated over the ocean leads to a regional decrease in SST south of the equator with
an averaged reduction of about ∼-0.3 to 0.5 K during the JJA season (Figure 4b) and a weaker impact beyond 15°W. It should
be noted that this SST-effect of the BBA estimated in the CNRM-CM simulations is consistent with the results obtained by
Solmon et al. (2021), but smaller in magnitude. This could be due to the slab ocean model vs. 3D oceanic model and to the
different magnitude of the low cloud response in the different modelling systems.

In addition to the SST cooling, and as shown in Figure 2b, BBA transported over the SEA generates additional solar heating
of ∼+0.5k per day between 600 and 800 hPa due to solar absorption. Both the local heating and the SST decrease contribute
to increasing the LTS (Figure 4c). Indeed, the CNRM-CM simulations indicate an important influence of the BBA radiative
forcing on the LTS over most of the tropical ocean below the equator. The results indicate a strengthening from about ∼+0.7 K
near the African coast to ∼+0.2 K up to 15°W. As mentioned previously, this positive impact in the LTS is partly responsible for
the increase in the low-cloud fraction by promoting the stabilisation of the lower troposphere below the aerosol layer (Figures
A3a,b). Indeed, the results indicate a positive anomaly (increase of the subsidence) of the vertical velocity (+0.01 Pa s$^{-1}$) over
the southeast Atlantic especially at 925 hPa. This impact can also be seen in the anomaly of the surface wind amplitude which
is reduced by about ∼-0.3 to -1 m.s$^{-1}$ over SEA (Figure A4 in Appendix).

This "stabilisation effect" of the lower troposphere is found to be associated with a decrease in the sinking at 700 hPa of
about ∼-0.01 Pa s$^{-1}$ between the African coast and ∼10°W (Figures 5a and A3f in Appendix). This is therefore associated
to a relative destabilisation in the vicinity of the maximum BBA heating rate (at 700 hPa) resulting in a relative decrease
of subsidence (Figure 5a). These BBA effects are mainly observed between the African coast and ∼10°W where the BBA
extinction coefficient (at 550 nm) is greater than 0.025 km$^{-1}$. For longitudes beyond, the impact on the vertical velocity is
small (Figure 5a). This additional buoyancy generated by BBA heating at these altitudes reduces the descent fluxes above the
cloud top, thereby limiting the intrusion of dry air from the free troposphere into the marine boundary layer and contributes
to the moistening of the marine boundary layer. Figure 5b clearly shows that the mass fraction of cloud liquid water within
the marine boundary layer is increased (by ∼10$^{-5}$ kg kg$^{-1}$) in response to the two main identified processes, explaining the
positive impact on the liquid water path and cloud optical depth simulated on the ocean below the equator (Figures 3b and 3d),
as discussed previously.

This positive impact of BBA on the low cloud fraction presents some differences with previous modeling studies using both
SST-forced or slab-ocean models (Sakaeda et al., 2011; Allen et al., 2019; Mallet et al., 2020 and Solmon et al., 2021). Even
if the comparisons are obviously not direct due to the differences in the configuration of the models used, spatial resolutions
or the representation of the clouds, comparisons with SST-forced simulations indicate discrepancies especially over the Gulf
of Guinea, where Allen et al. (2019) or Mallet et al. (2020) show a LCF decrease in opposite sign to the response obtained
with the CNRM-CM model. For these two studies, the amplitude of the BBA effect on the low cloud fraction is also found to
be smaller than in the coupled CNRM-CM simulations. In opposite, the modeling studies using a slab-ocean model (Sakaeda





et al., 2011; Mallet et al. 2019) show better agreement with the CNRM-CM model, with a predominantly positive effect over

the ocean. However, the amplitude of the low-cloud fraction response is found to be stronger in the coupled model than in the
RegCM-SOM model, with an impact of up to 15°W in the CNRM-CM simulations.

   Finally, one of the original features of these new ocean-atmosphere coupled simulations concerns the use of the 3D ocean
model NEMO. This allows, for the first time to our knowledge, to study the way in which ocean cooling propagates at depth in
addition to the SST response. Figure 4d clearly shows that the ocean temperature is largely influenced during the JJA season
to a depth of about ∼50-60 m between the coast and 10°W. The CNRM-CM simulations indicate a homogeneous cooling
between the surface and about ∼20 m depth, with a temperature anomaly averaging ∼0.3K. Figure 4d shows a second cooling
zone at greater depths ∼40 to 60 m, where the negative anomaly may exceed that identified between the surface and 20 m. The
results also indicate that the cold anomaly propagates to depth of about 80 m during this season. Simulations indicate also an
impact on the upper ocean dynamics (Figure 4e) where the model simulates a westward equatorial current (-0.2 to -0.3 m.s$^{-1}$)
associated with a counter-current off the coast of West Africa.

   In response to the BBA anomaly, the zonal equatorial surface current presents an eastward anomaly which means a slowing
in absolute velocity by 0.05 to 0.1 m.s$^{-1}$ from the Atlantic coast to 20°W. This "slowing" effect appears to be more intense
and spatially more widespread westwards (up to 30°W) north of the equator than the simulated impact south of the equator,
which is mainly confined between the African coast and 15°W. In parallel, the eastward counter-current further north is also
slightly weakened, certainly in response to the slowing of the flow at the equator, although the effect is found to be more limited
(-0.03 m.s$^{-1}$). The most pronounced impact on the counter-current is simulated along the coasts of the Gulf of Guinea, with
an effect that can exceed -0.05 m.s$^{-1}$. As shown in Figure 4f, this slowing of the surface zonal current along the Equator may
be due to the surface wind anomaly induced by the radiative effect of BBA. Indeed, the cooling of the SST and the continental
temperature near the African coasts, especially Angola, generates an anticyclonic anomaly (not shown) that favours important
changes in wind direction with a more pronounced north-easterly component near the equator (Figure 4f). This wind anomaly
at the surface may partly explain the slowing of the surface current as simulated in the CNRM-CM model. However, a more
detailed analysis of these results related to the change in oceanic temperature and surface dynamic will be carried out in a
complementary study.

### 3.2.2   Changes caused by BBA on Central and Western Africa

Over land, Figure 3a shows different regional responses of low cloud fraction, with impacts mostly positive over Central
Africa and slightly negative over western Africa. Over Central Africa, the CNRM-CM simulations show an increase of low
cloud fraction (by more than ∼+10%) over Angola and the coastal region of Gabon, associated with more moderate (∼+2 to
5%) changes over the Democratic Republic of Congo (DRC). The low cloud response simulated in the coastal areas of Gabon
and Angola is likely due to the coupling between the ocean and the atmosphere being taken into account, and in particular
the response (decrease) of the SST near the coast (Figure 4b), which accentuates the positive surface pressure anomaly (not
shown). The impact on the cloud fraction in the coastal zones of Angola and Gabon appears to be more pronounced in coupled
simulations than in studies using an SST-forced or slab ocean model approach (Sakaeda et al., 2011; Mallet et al., 2019). In



parallel and over Central Africa, the decrease in surface air temperature over the continent (Figure 4a), in addition to diabatic heating, leads to the stratification of the lower troposphere and limits convection (Figure 6b), which helps to maintain low

cloudiness. In addition to the low-cloud fraction, these new coupled simulations also show an effect on the water content and optical properties of the clouds in Central Africa, increasing the liquid water path and cloud optical depth over Gabon and Angola by about ∼0.01 kg m$^{-2}$ and ∼2, respectively. This contributes also to the sharp reduction in solar radiation at the surface simulated over these regions (Figure 3c).

In addition to the low-level cloud analysis, Figure 6a shows the BBA impact on mid-level clouds for the JJA period. The

results generally show an opposite effect to that obtained for the low-level clouds, with a decrease (by ∼5-10%) in the cloud fraction especially over Central Africa. Over this region, the main processes involved are identical to those discussed previously, namely a decrease in the continental surface temperature associated with radiative heating of the lower troposphere induced by smoke aerosols. Indeed, the significant BBA surface radiative forcing simulated in JJA (-20/-30 W.m$^{-2}$, Figure 2a) leads to a large decrease in the surface temperature of about -1 to -1.5 K (Figure 4a) over a large part of Central Africa. This surface

cooling, combined with the additional heating due to BBA, induced a more stable lower troposphere below the BBA plumes. As mentioned previously, such effects reduce the vertical ascent and convection, especially between the surface and 700 hPa (Figures 6b and A3). At such altitudes, the vertical velocity anomaly is about +0.01 to 0.02 Pa.s$^{-1}$ reflecting a reduction of the average convection over the continent especially between 5 and 15°S. As shown in Figure 6c, the combined effects explain the decrease in mid-level cloud fraction for atmospheric levels above ∼700 hPa over Central Africa, with a well-marked reduction

up to 400 hPa (for latitudes between 5 and 15°S).

Over the central African region, the competing effects between the positive anomaly of the low-cloud fraction (Figures 3a and 6c) and the mass fraction of cloud liquid water (Figure 5b), and the negative anomaly for the same variables (Figures 5b, 6a,c) with respect to mid-level clouds, lead to a decrease in the cloud optical depth of the order of ∼-1 (Figure 3d). Despite this decrease, the BBA direct effect is found to be predominant over this region and explains the significant solar dimming over

central Africa (Figure 3c). Finally and with regard to high-level clouds, this study shows little impacts of the BBA radiative effect on the cloud fraction (Figure 6c).

### 3.3 How do the BBA affect the lower troposphere dynamic and precipitation in JJA ?

The impact of the BBA radiative effect on the lower troposphere dynamics has been analysed using the wind field anomalies at 925 and 850 hPa (Figures 7). Near the surface, the results show moderate northerly wind anomalies along the Angolan coast

and the Gulf of Guinea. In addition, Figures 7 shows a cyclonic anomaly at 850 hPa generated by the BBA radiative effects and solar heating. The BBA radiative feedback implies wind changes mostly between 15°E and 15°W with north-westerly anomalies over the ocean south of the Equator combined to more northerly anomalies (of ∼1 m s$^{-1}$) along the Angolan coast. Hence and for both heights, the simulated wind anomalies show a general northwesterly flow over the Gulf of Guinea, which opposes to the West African monsoon normal flow. As discussed hereafter, the creation of this atmospheric dynamic anomaly

at 850 hPa has potentially an impact on the advection of moisture towards central and southern Africa with implications for rainfall in JJA. Compared to SST-forced simulations, the meridional anomaly simulated at the surface along the Angolan coast





($\sim$0.5-0.6 m s$^{-1}$) in the CNRM-CM model is found to be higher than reported in Mallet et al. (2020) with an anomaly of $\sim$0.3 m s$^{-1}$. On the contrary, the response is much more consistent in terms of amplitude with the RegCM-SOM simulations (Mallet et al., 2020), which seems to indicate that the coupling with the ocean modulate the BBA impact on the low-level atmospheric dynamic by enhancing the response of the circulation.

As mentioned in the introduction, the hydrological sensitivity to the radiative forcing of absorbing aerosols is very large due to the high complexity of the interactions. The response of precipitation to the forcing of absorbing particles is particularly uncertain over the tropics (Samset et al., 2016, 2022; Persad, 2023) as such aerosols can exert a rapid adjustment of precipitation in addition to the feedback from surface temperature change (referred to as the 'fast' and 'slow' response, respectively). Figure 8a shows the averaged JJA anomaly (in mm per day) in total precipitation simulated for the period 1990-2014. The largest negative signal is obtained over the West African coast with a decrease of about $\sim$-1 to -1.5 mm per day over the coastal regions of Guinea, Sierra Leone, Ivory Coast, Liberia and Nigeria. On the other hand, the CNRM-CM simulations reveal a moderate positive impact over northern Angola. Over this region, the positive effect on the JJA precipitation (increase of $\sim$+0.5 mm per day) is found to be less pronounced than the drying impact simulated over the coastal Western Africa.

The impact on precipitation over the West African coast and part of the Gulf of Guinea is consistent with the local shift of the convective rainbelt towards the south near the equator as shown in Figure 8a (in response to the radiative forcing and solar heating exerted in the outflow), and associated with the 925 and 850 hPa wind anomalies (Figures 7). This change in the atmospheric dynamics of the lower troposphere tends to decrease relative humidity over the West African coast and increase it south of the Gulf of Guinea (Figure A4a), which may explain the negative response of precipitation. This drying effect over the West African coast is also shown in the convective condensed water path anomaly, indicating a decrease from -0.002 to -0.005 kg.ms$^{-2}$ over these regions (Figure 8b). The BBA effect on precipitation simulated by the CNRM-CM model over the oceanic tropical region is found to be consistent with the results of Solmon et al. (2021), who also indicated a general decrease in precipitation (by $\sim$-1.5 to 2 mm per day) over this region. However and although the overall rainfall response (drying effect) is consistent between the two modelling systems, there are some discrepancies in the regional response, in particular the drying over the coastal regions of Liberia, Sierra Leone and Guinea which is more important in this study compared to the RegCM simulations. On the other hand, the reduction in rainfall simulated by the CNRM-CM model in the Gulf of Guinea is less pronounced compared to Solmon et al. (2021). Finally, the hydrological response is opposite over the Central African continent, where the RegCM model simulates mainly a decrease in precipitation unlike the CNRM-CM model over northern Angola.

Indeed and as mentioned previously, the CNRM-CM simulations also show a moderate positive ($\sim$+0.3 mm per day) impact on precipitation over northern Angola (Figure 8a) in agreement with the relative and specific humidity positive anomalies simulated over this region (Fig. A5 in Appendix). As discussed previously and as shown in Figure A3 (Appendix), BBA induce a lower tropospheric stratification associated with positive vertical velocity anomalies at both 850 and 700 hPa ($\sim$+0.02 Pa.s$^{-1}$) that would a priori promote a reduction in the convective rainfall, which is contrary to the positive anomaly simulated in this region. Over Central Africa, the moisture advection response to BBA forcings seems to play a role in the moderate increased precipitation. As mentioned above, BBA tend to produce a cyclonic anomaly at 850 hPa over the ocean (Figure 7b),



which then translates into a north-westerly wind anomaly near the coasts of Angola, which is likely to transport moisture inland
(Figure A5) more efficiently then favoring precipitation. This hypothesis is supported by the analysis of the vertical profile of
the specific humidity tendency anomaly (averaged over the box 15E-25E/0-15S) for the two terms related to advection and
convection (Figure A6). The results clearly show a positive anomaly above $\sim$850 hPa (from 2 to 3 $10^{-9}$ kg.kg.s$^{-1}$) for the
advection tendency term while the anomaly for the convective term is found to be smaller. Hence and as shown for the western
African region, the rainfall anomaly observed over Angola seems to be related to the response of atmospheric dynamics and
an additional supply of moisture from the Atlantic ocean.

## 3.4    Response of the tropical African climate to BBA emissions during the SON season

In this part, the analysis focuses on the temporal propagation of the BBA radiative effect for the SON season. This analysis is
motivated by the fact that biomass burning emissions, although lower than during JJA, are still present in September-October
(Redemann et al., 2021) and would have residual impacts. The interactions between BBA and the tropical African climate are
potentially important during this season as SON is characterised by the second phase of the West African monsoon (southward
retreat of the rain band from the Sahel to the Guinean coast), with precipitation occurring mainly over the coastal regions
(around $\sim$30 to 40 mm per month) with maxima in October (Maranan et al., 2018).

As shown in Figure A7 (Appendix), the BBA AOD anomaly simulated by the CNRM-CM model during SON is logically
lower than during JJA, consistently with the decrease in biomass burning emissions. The AOD reaches $\sim$0.3-0.4 (at 550 nm)
over the African continent and decreases rapidly to $\sim$0.05-0.1 over the Atlantic. In contrast to JJA, the SON AOD anomaly
results in a direct surface forcing that is mainly localised over central Africa ($\sim$-5 W m$^{-2}$, Figure A7) with a weak radiative
forcing over the ocean, except near the African coast. As for JJA, the simulations show a positive effect of BBA on the low-
level cloud fraction with an increase of $\sim$5 % over the entire tropical ocean and a detectable effect as far as the coast of
Brazil (Figure 9a). Although the impact near the coast of Africa is less pronounced than in JJA, the response is much more
widespread during the SON and the effect is remarkable between the equator and 15°S, even though the amount of BBA is
lower and mainly restricted between the sources and the African coast. The results even indicate the existence of a gradient in
the low-cloud response across the ocean basin, with a more pronounced effect between 15°W and the coasts of Brazil compared
to the response simulated on SEA. Contrary to JJA, Figure 9a also shows that the low cloud fraction over the coastal areas of
Angola and Gabon is not influenced by biomass burning emissions during this season. In terms of cloud liquid water path,
Figure 9b also shows an increase by about $\sim$0.005 kg.m$^{-2}$ over most of the oceanic region, except over the Gulf of Guinea
where the simulations indicate a moderate decrease. As noted for the low-cloud fraction, the results clearly show a significant
spatial shift compared to the JJA and a more pronounced effect to the west of the tropical ocean, with a maximum anomaly
(0.01 kg m$^{-2}$) at about 30°W. The overall effect is to increase the liquid cloud optical depth by $\sim$1 over the entire oceanic
region south of the equator (Figure 9c), an effect that is less pronounced than during JJA (Figure 3b).

The results also show important changes in the cloud properties over part of Brazil, but the present simulations do not allow
to disentangle the contribution of South American emissions from a possible effect of Central African emissions. Although the
different impacts are likely attributed to the BBA emitted over South America in SON (Figures A1 and A7), the contribution



of African emissions is not necessarily negligible along the eastern coast of South America. Indeed, as recently reported by Holanda et al. (2023), African smoke could influence aerosol-radiation interactions over the Amazon, with the strongest effect on the eastern basin. This potential impact of BBA over South America will be the subject of a specific study using dedicated simulations.

In a manner consistent with the BBA AOD and the induced changes in tropical cloud properties, the results indicate important effects on the surface solar radiation during SON. Figure 9d shows a decrease near biomass burning sources ($\sim$-15 W.m$^{-2}$), combined with a large impact on the ocean south of the Equator. Over the entire oceanic region, from Africa to the coast of Brazil, solar radiation is reduced by $\sim$-10 to -15 W m$^{-2}$. As shown in Figures 9a,b,c and A7, this dimming is largely due to the increase of the low-level CF, LWP and COD, while the BBA direct effect more likely plays a smaller role in this season.

In parallel to the changes in clouds and radiation, the results reveal a persistent impact on SST that is still present during SON in the coupled simulations, with a decrease of about -0.5 K over a large part of the tropical ocean (Figure 10b). In contrast to JJA, the effect is widespread over much of the region, with a decrease simulated over the entire tropical ocean between the equator and 15°S. The simulations show that the SST is largely affected between 0° and 30°W, highlighting an influence on a much larger spatial scale than during the period of maximum smoke emission. This SST change is likely due to the strong inertia in the ocean temperature response due to the BBA direct forcing and cloud feedback. Recently, Solmon et al. (2021)

have also indicated a decrease in SST between -0.4 and -0.7 K at $\sim$5-10°S but in contrast to the CNRM-CM simulations, they report a more pronounced time lag between the BBA radiative forcing maximum (occurring in August/September) and the maximum in the effects on SST (occurring in October/November). Differences in the low-level cloud response and the ocean parameterisation (oceanic model vs. Slab Ocean Model) may explain the variability in the time evolution of the SST response.

The positive feedback on the low-level cloud fraction and liquid water path simulated in SON over the ocean (Figures 9a,b) is mainly controlled by this persistent SST cooling. Indeed, the large spatial extension of the cloud response cannot be explained by the BBA forcing itself which is weaker and mainly confined between the coasts of Africa and $\sim$5° East in SON (Fig. A7 in Appendix). As described for the JJA season, the decrease in SST is likely at the origin of the increase in LTS (by about $\sim$0.5 K, Figure 10c) and consequently in the low-level cloud fraction and the integrated water content, indicating that the "slow

response" is also a crucial process in the SON season. In parallel and as for the JJA season, the simulations show, for the first time to our knowledge, that the ocean cooling is propagated at depth in SON, notably between 0° and 30°W (Figure 10d). Indeed, the coupled simulations indicate that oceanic temperatures are reduced by $\sim$-0.3 down to 100 m for latitudes between 0 to 15°S. In addition to the cooling, a positive temperature anomaly is also simulated between the coast and $\sim$5°E up between $\sim$30 and $\sim$80m depth, that could possibly be due to changes in the ocean dynamics. As mentioned above, a specific study will

be carried out to analyse the dynamical response and the oceanic temperature changes.

Finally, Figure 11 shows that the overall impact of BBA radiative forcing on the SON precipitation is mainly observed over the ocean, with no major changes over the continent. The major impact is localized in the Gulf of Guinea with a decrease in precipitation by $\sim$-1 mm by day. In contrast to JJA, this reduction is more related to the impact of BBA on the SST in SON, which limits convection on the Gulf of Guinea and precipitation. This inhibition is characterised by a reduction of the

vertical velocity (Figure A8 in the appendix) over this region by about $\sim$-0.005 to -0.01 Pa.s$^{-1}$ at 850 and 700 hPa. As shown





in Figures 11 and A8, the maximum reduction in vertical velocity over the Gulf of Guinea corresponds to the regions where precipitation is most affected.

## 4    Conclusions

The impact of BBA emitted in central Africa on the tropical African climate is studied using the CNRM-CM model in the
ocean-atmosphere coupled configuration including an interactive representation of smoke aerosols. The BBA direct and semi-direct effects on clouds, surface solar radiation, surface-atmosphere flux exchange, atmospheric dynamics and precipitation are analysed first during the JJA season (period 1990 to 2014). This study indicate an important impact of the BBA radiative effects on the low-level clouds with an increase in the cloud fraction, water content and optical depth of the order of $\sim$5-10%, $\sim$0.01 kg m$^{-1}$ and $\sim$1-2 over the southeast Atlantic, respectively. Compared to previous studies using global or regional models in a
SST-forced configuration, the effect of BBA on low-level cloud fraction simulated by the coupled CNRM-CM model is found to be more homogeneous and more intense with a positive feedback simulated over the whole Atlantic ocean. This positive impact is found to be mainly due to the SST decrease, in response to the surface BBA radiative forcing ($\sim$-5 to -15 W.m$^{-2}$) and cloud changes, associated to lower tropospheric heating, that contribute to (*i*) increasing the LTS and (*ii*) to limiting the intrusion of dry air at the cloud top.

Over land, this study indicates a positive effect of BBA on the low cloud fraction, liquid water path and optical depth for coastal regions of Gabon and Angola with a potentially enhanced effect in coupled simulations that integrate the response (cooling) of the SST. In addition, the impact of BBA on the continental mid-level clouds is also found to be important during the JJA season, with a reduction of about 10% over Central Africa, mostly due to BBA radiative effects inhibiting the convection over the continent. In terms of atmospheric dynamical perturbations, the simulated northwesterly anomalies of winds near the
surface and at 850 hPa are found to weaken the West African monsoon flow. The results obtained from these new coupled simulations indicate that changes in the lower troposphere dynamics impact the precipitation in JJA with negative feedback over the coastal regions of Liberia, Sierra Leone and Guinea (reduction of $\sim$-1 to 1.5 mm per day), associated to moderate increase over northern Angola ($\sim$+0.3 mm per day).

Another important result is that BBA emissions are shown to have an important effect during the SON season, which is
characterised by the second phase of the West African monsoon. Indeed and despite lower smoke emissions, some large impacts in cloud properties and radiative budget are still present in SON, which is likely the signature of the "slow" SST feedback. In particular, the coupled simulations highlight a persistent effect on low-level clouds (increasing both CF, LWP and COD) over a large part of the Atlantic ocean especially above 15°W, associated to precipitation decrease ($\sim$-1 mm per day) over the Gulf of Guinea.

Interestingly and for the first time to our knowledge, these new coupled CNRM-CM simulations that use a full ocean model (NEMO) have highlighted an impact of BBA radiative effects on the tropical ocean temperature for both studied seasons. The results show that temperatures could be decreased at depths of up to $\sim$50-80 m depending on the analysed seasons. Simulations indicate different regional extent of the temperature responses with more limited impact in JJA, which nevertheless corresponds



to the maximum of biomass-burning emissions. During the SON season, the cooling is simulated at depth of 80 m up to
approximately 30°W but the general decrease in the ocean temperature is associated to a positive anomaly near the African
coast (below 30 m). In parallel to the ocean temperature, the zonal equatorial surface current presents an eastward anomaly
which means a slowing in absolute velocity (by 0.05 to 0.1 m.s$^{-1}$) from the Atlantic coast to 20°W. This slowing of the surface
zonal current along the Equator may be due to the surface wind anomaly induced by the radiative effect of BBA. Following
these original initial results, a complementary study will be carried out to look more specifically at the response of the ocean
and in particular at the surface currents, density of waters and 3-D circulation for both seasons.

Recently, numerous studies have clearly demonstrated the important role of the rapid response of the tropical precipitation
to absorbing aerosols. The present study confirms such results but also highlights that the slow SST response over this tropical
region must be considered, as it contributes significantly to the modification of cloud properties, surface radiations, atmospheric
dynamics and precipitation. In addition, some of the BBA effects initiated during the dry season can then be propagated to the
SON season mainly due to the inertia of the ocean (cooling) temperature responses. In this sense, the use of coupled models
seems to be essential to address the different impacts of BBA over this tropical region during the period corresponding to the
peak of smoke emissions, but also during other seasons of the year. For example, these new CNRM-CM simulations could also
be used to investigate the role of winter emissions that are further north than the study region, but which can be transported
across the Gulf of Guinea and affect the SST.

The proposed method is a first step towards future modelling exercises that can be improved. First, the present study is only
based on the CNRM-CM simulations and could be carried out using different large-scale coupled models, with fixed biomass-
burning emissions and absorbing optical properties, in order to consolidate the results and better quantify uncertainties. Second,
new simulations need to be carried out using a version of the CNRM-CM model that better resolves the low-level clouds over
this region. Indeed, this global model, as most of GCMs, presents a bias in the representation of these clouds, which can limit
the reflection of solar radiation and possibly lead to underestimate the BBA radiative heating. As this parameter is crucial for
the response of low-level clouds and the atmospheric dynamics over this region, it is possible that the impacts and feedback
highlighted in this study would be more pronounced in a version of the model that correctly represents the stratocumulus
clouds.

Finally and in order to improve the proposed protocol, additional coupled simulations will be carried out that independently
consider only specific tropical emissions (from the Amazon and Indonesia) which could potentially affect this region. These
will allow a more detailed analysis of the possible response of tropical Africa's climate to smoke emissions from other regions.
In addition, new CNRM-CM simulations will be carried out to analyse the sensitivity of the regional climate response to the
absorbing properties of BBA. Finally, the integration of desert dust in the future simulations is envisioned to analyse the joint
response of the direct and semi-direct radiative impacts of the mixed aerosols on the tropical African climate.





**Acknowledgments**

We are grateful to the entire group in charge of the CNRM climate models provided support to us. Supercomputing time was provided by the Météo-France/DSI supercomputing center.

**Data availability**

This study relies entirely on publicly available data. All the output files from ARPEGE-Climat simulations used in the present study are available on requests.

**Author contributions**

MM, AV, PN and RR designed the modeling study and developed the analysis protocols. FS and TD contributed to the data
analysis. All authors reviewed the final manuscript.

**Competing interests**

The contact author has declared that none of the authors has any competing interests.



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





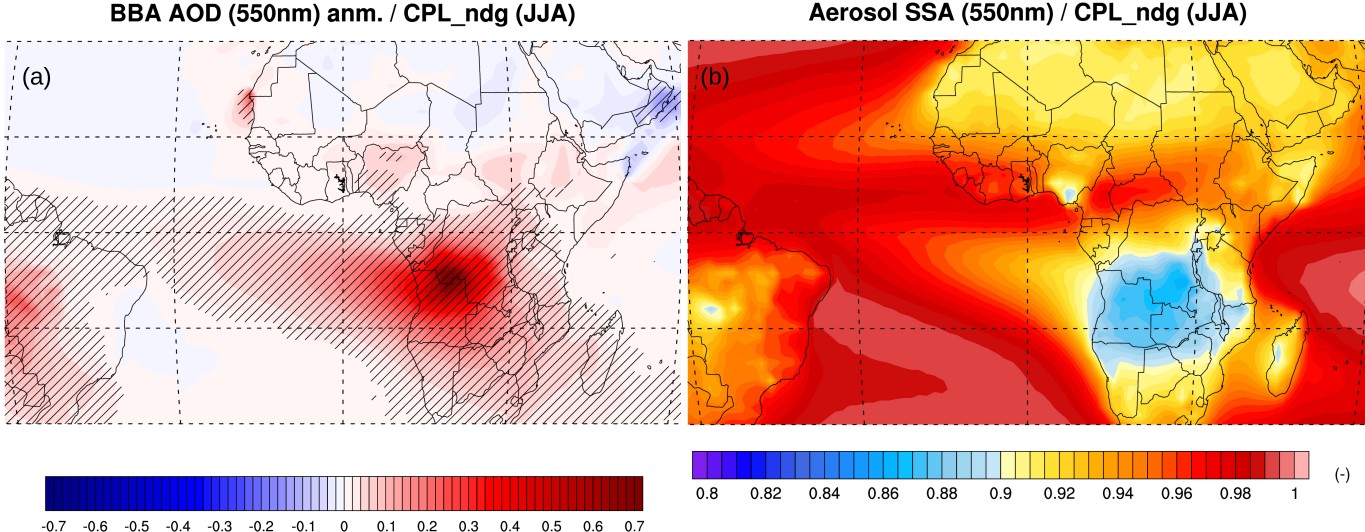

**Figure 1.** Averaged (1990-2014) seasonal (June-July-August) of a) anomaly of BBA Optical Depth (at 550 nm) and b) effective Single Scattering Albedo, SSA (at 550 nm) simulated by the CNRM-CM model.





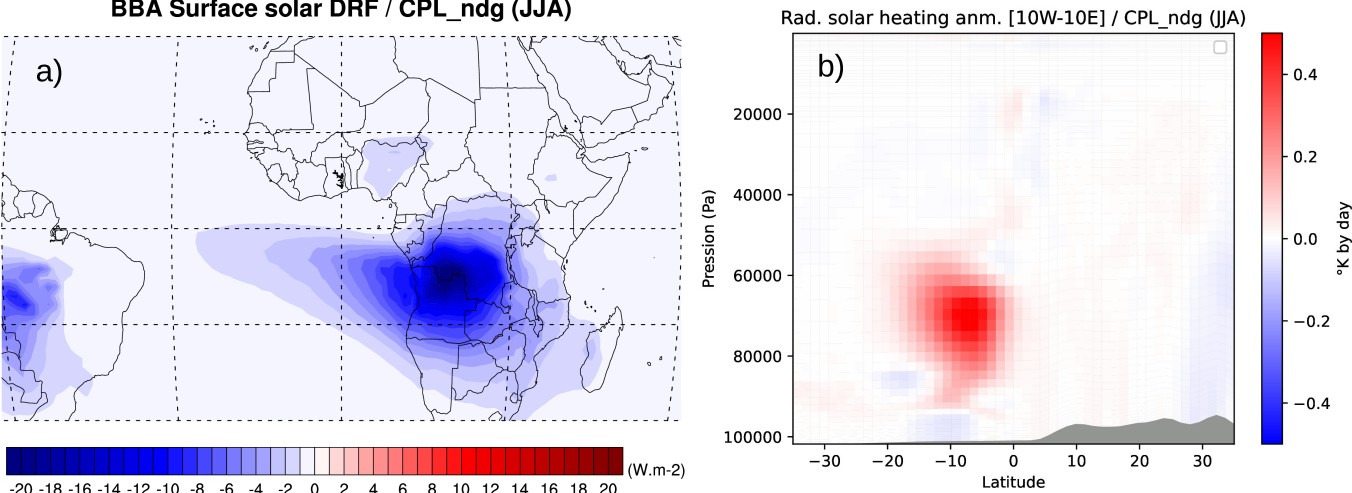

**Figure 2.** Averaged (1990-2014) seasonal (JJA) BBA a) surface direct radiative forcing (in W m$^{-2}$) and b) anomaly of solar heating rate along the latitude (in K by day and averaged between 10°W and 10°E).



**Figure 3.** Averaged (1990-2014) seasonal JJA anomaly of a) low cloud fraction (in %), b) liquid cloud Water Path (in kg.m$^{-2}$), c) surface downward solar radiations (in W.m$^{-2}$) and d) cloud optical depth, simulated by the CNRM-CM model. Hatching indicates regions with a significant effect at the 0.05 level (Wilks test).





**Figure 4.** Averaged (1990-2014) seasonal JJA anomaly of a) air surface temperature (in K), b) sea surface temperature (SST, in K), c) Lower Tropospheric Stability (LTS, in K, defined between the surface and 700 hPa), d) vertical profile of the oceanic temperature along the longitudinal transect from 15°E to 40°W (averaged between 0 and 15°S, in K, the isolines represent the climatological values), e) surface zonal oceanic current (in m s$^{-1}$, the isolines represent the climatological values) and f) the surface wind speed. For the figures a,b,c,d, the hatching indicates regions with a significant effect at the 0.05 level (Wilks test). For the figures e and f, anomalies are only plotted for regions where the statistical test is respected.



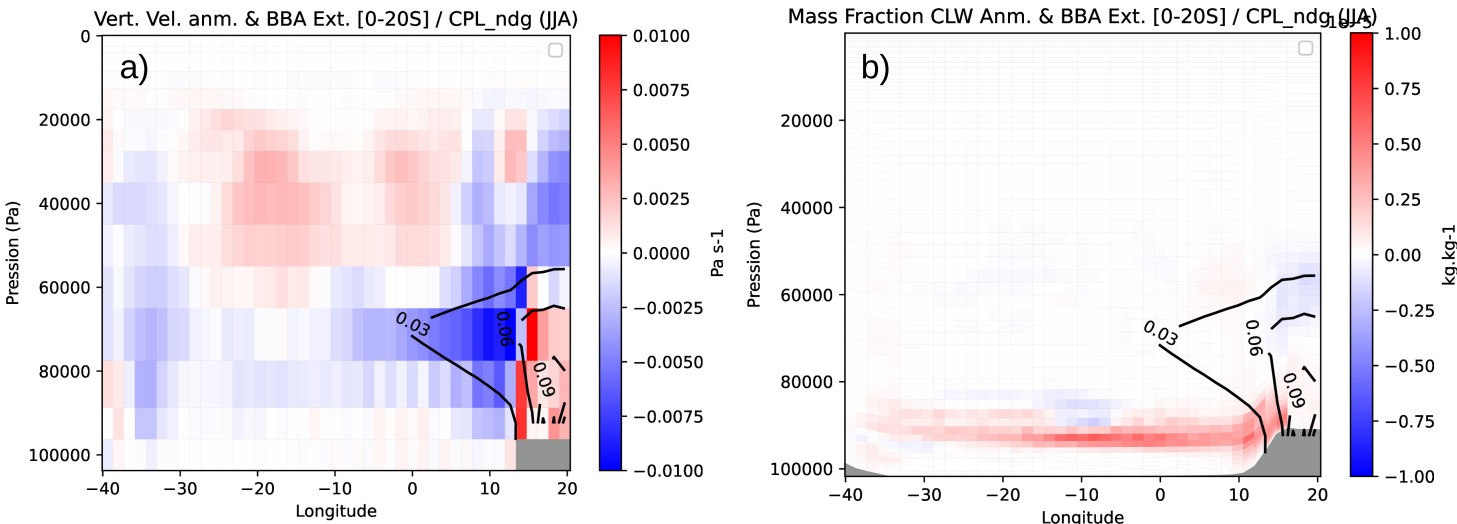

**Figure 5.** Averaged (1990-2014) seasonal JJA anomaly in the vertical velocity vertical profiles along the longitudinal transect from 20°E to 40°W (in Pa s$^{-1}$, left) and mass fraction of Cloud Liquid Water vertical profiles for similar longitudinal transect (in kg kg$^{-1}$, right). Both variables are averaged between 0 and 20°S and the isolines reported on each transects represent the smoke extinction coefficient at 550 nm (in km$^{-1}$)





**Figure 6.** Averaged (1990-2014) seasonal JJA anomaly of a) medium-cloud fraction (in %), b) latitudinal transect from 5°N to 20°S of cloud fraction vertical profiles (averaged between 15-25°E, in %) and c) latitudinal transect of vertical velocity profiles from 5°N to 20°S (averaged between 15-25°E, in Pa.s$^{-1}$).





a) 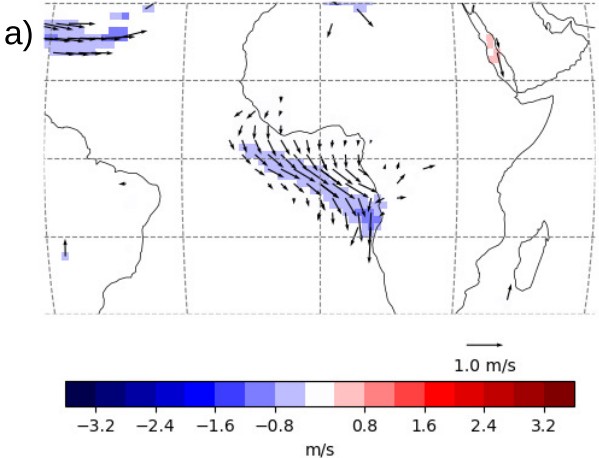

b) 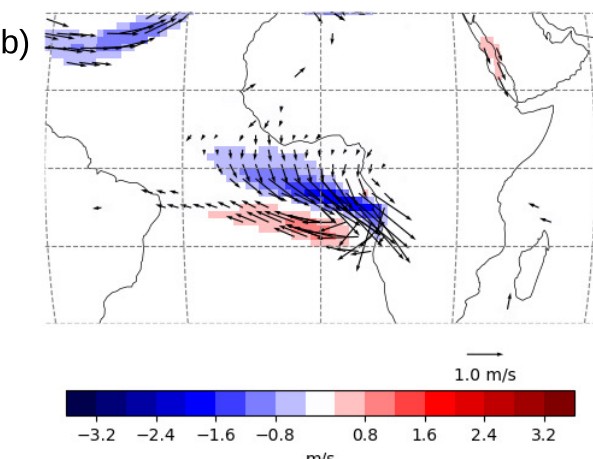

**Figure 7.** Averaged (1990-2014) seasonal JJA anomalies in the wind speed at 925 and 850 hPa (in m s$^{-1}$). The significance of the wind field change in both intensity and direction is reported (arrows reported for zones characterised by weak shading indicate that the significance of the wind field change is mainly related to a rotation of the wind.



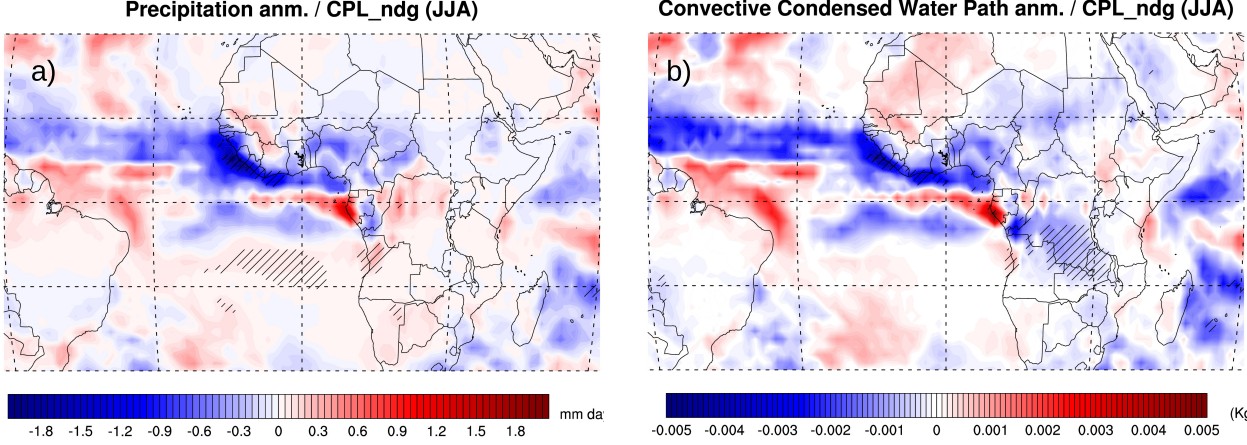

**Figure 8.** Averaged (1990-2014) seasonal JJA anomalies in a) the precipitation (in mm by day) and b) convected Condensed Water Path (in kg.m$^{-2}$). Hatching indicates regions with a significant effect at the 0.05 level (Wilks test).

**Low Cloud Fraction anm. / CPL_ndg (SON)**

**Liquid Water Path anm. / CPL_ndg (SON)**

**Liquid Cloud Optical Depth anm. / CPL_ndg (SON)**

**SW Radiation Surface anm. / CPL_ndg (SON)**

**Figure 9.** Averaged (1990-2014) seasonal September-October-November (SON) anomaly of a) Low Cloud Fraction (in %), b) Liquid Water Path (in kg.m$^{-2}$), c) liquid cloud optical depth (no unit) and d) downward surface solar radiations (in W.m$^{-2}$). Hatching indicates regions with a significant effect at the 0.05 level (Wilks test).



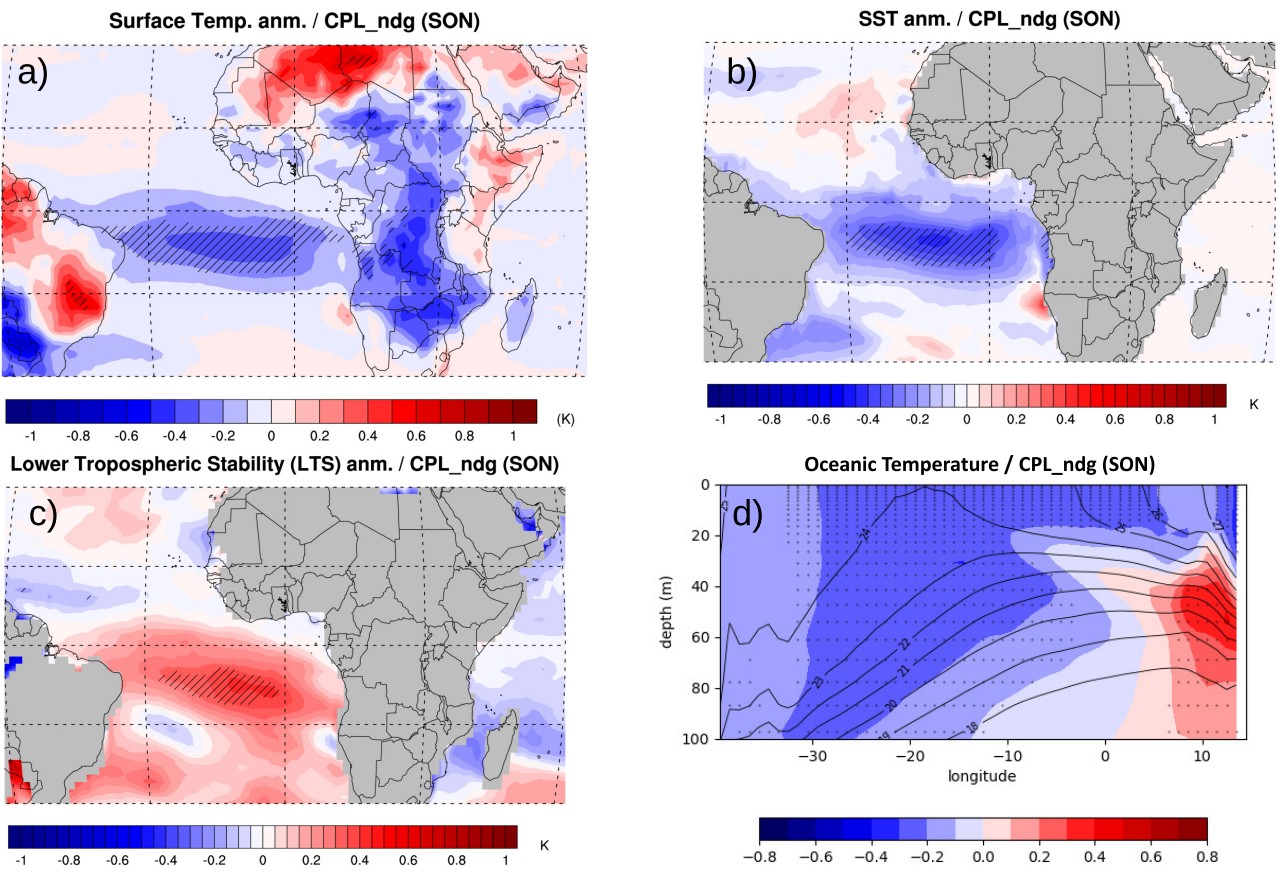

**Figure 10.** Averaged (1990-2014) seasonal SON anomaly of a) air surface temperature (in K), b) Sea Surface Temperature, SST (in K), c) Lower Tropospheric Stability, LTS (in K) and d) the oceanic temperature vertical profiles from 15°S to 40°W (averaged between 0 and 15°S, in K). Hatching indicates regions with a significant effect at the 0.05 level (Wilks test).



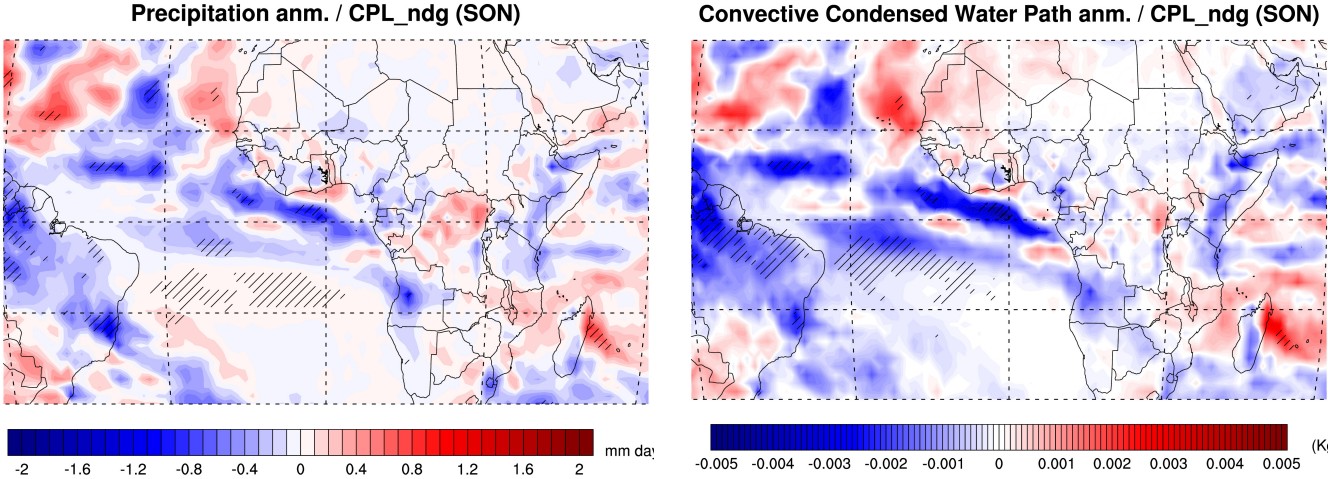

**Figure 11.** Averaged (1990-2014) seasonal SON anomalies in a) the total precipitation (in mm by day) and b) convected condensed Water Path (in kg.m$^{-2}$). Hatching indicates regions with a significant effect at the 0.05 level (Wilks test).



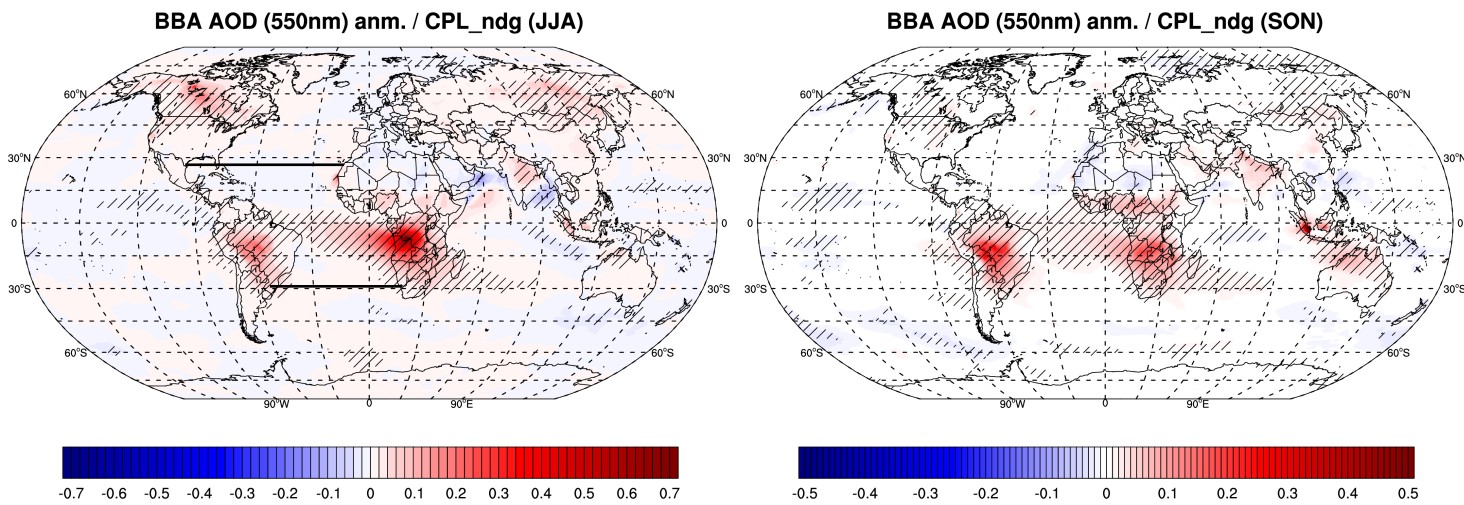

**Figure A1.** Averaged (1990-2014) anomaly of total AOD for the JJA (left) and SON (right) seasons simulated by the CNRM-CM model at the global scale. Hatching indicates regions with a significant effect at the 0.05 level (Wilks test). The latitudinal limits of the oceanic nudged domain are shown in the left figure.



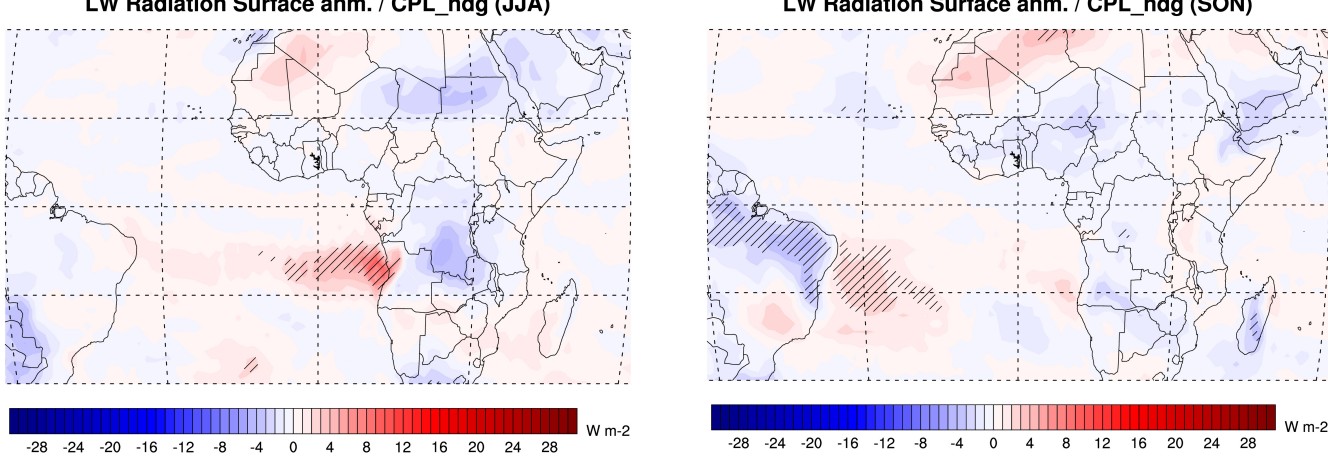

**Figure A2.** Averaged (1990-2014) anomaly of the surface longwave downward radiations for the JJA (left) and SON (right) seasons simulated by the CNRM-CM model. Hatching indicates regions with a significant effect at the 0.05 level (Wilks test).



**Figure A3.** Averaged (1990-2014) seasonal (JJA) vertical velocity (in $Pa.s^{-1}$) at different altitude levels 950, 850 and 700 hPa (left; a,c,e) and the corresponding anomalies (right; b,d,f). Hatching indicates regions with a significant effect at the 0.05 level (Wilks test).



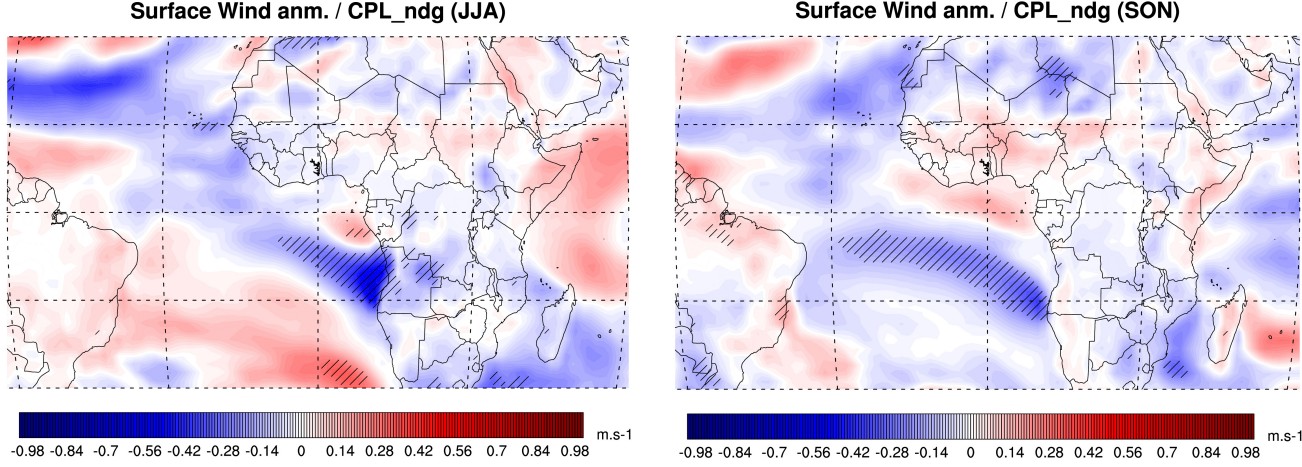

**Figure A4.** Averaged (1990-2014) anomaly of the surface wind (in m.s$^{-1}$) for the JJA (left) and SON (right) seasons simulated by the CNRM-CM model. Hatching indicates regions with a significant effect at the 0.05 level (Wilks test).



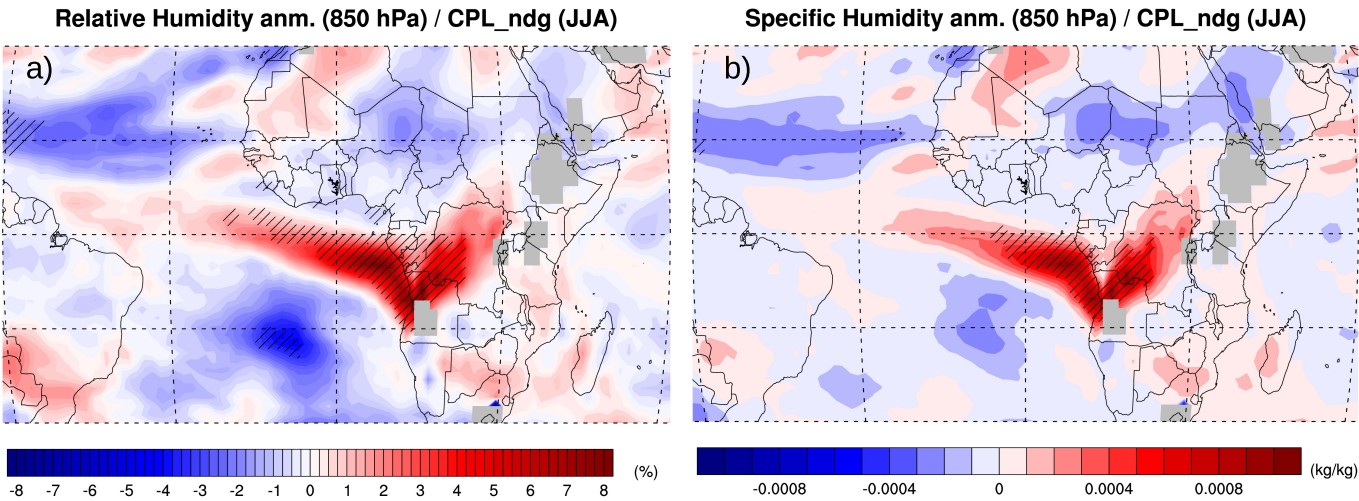

**Figure A5.** Averaged (1990-2014) seasonal (JJA) anomaly of the a) relative humidity (in %) and b) specific humidity (in kg/kg) (at 850 hPa) simulated by the CNRM-CM model. Hatching indicates regions with a significant effect at the 0.05 level (Wilks test).



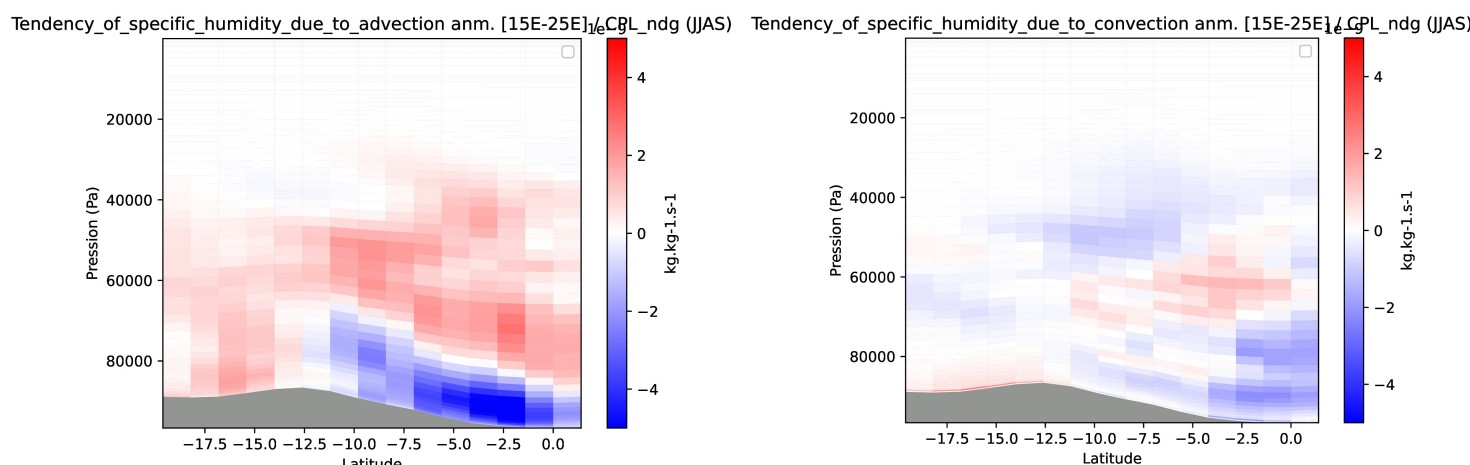

**Figure A6.** Averaged (1990-2014) seasonal (JJA) anomaly of the latitudinal transect, from 5°N to 20°S, of the specific humidity trends due to the advection (left) and convection (right) (in kg.kg$^{-1}$.s$^{-1}$, averaged between 15-25°E).



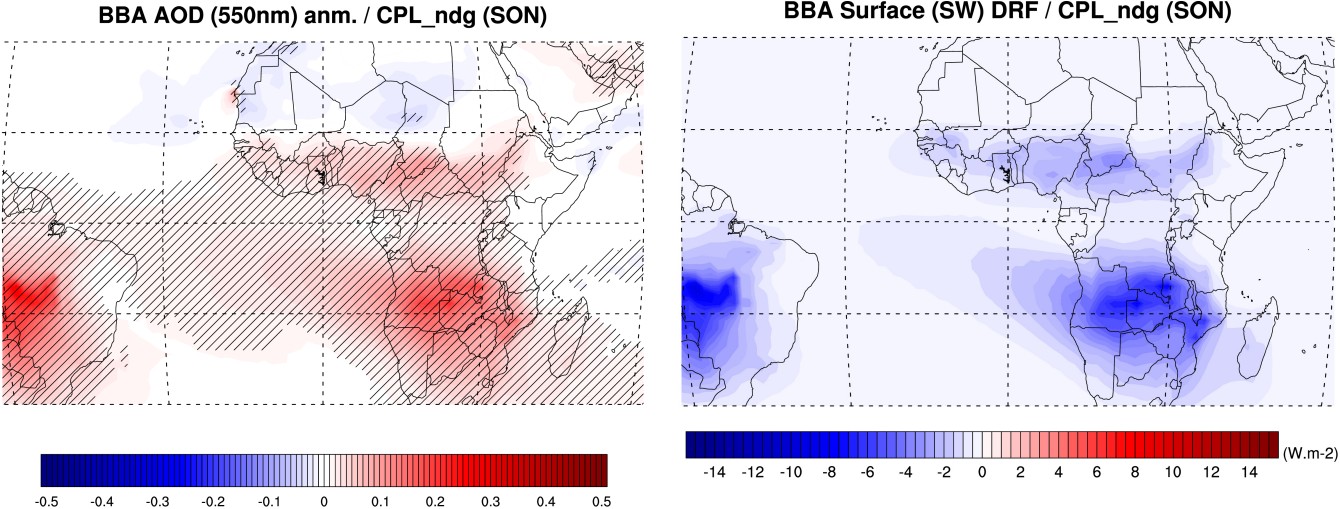

**Figure A7.** Averaged (1990-2014) seasonal (SON) anomaly of a) BBA Optical Depth (at 550 nm, hatching indicates regions with a significant effect at the 0.05 level (Wilks test)) and b) effective SW surface direct radiative forcing simulated by the CNRM-CM model.



**Figure A8.** Averaged (1990-2014) seasonal (SON) vertical velocity at the vertical levels of 850 and 700 hPa (left; a),c), in Pa.s$^{-1}$) and the corresponding anomalies (right; b),d), in Pa.s$^{-1}$) simulated by the CNRM-CM model. Hatching indicates regions with a significant effect at the 0.05 level (Wilks test).