# Peer review of "Impact of Biomass Burning Aerosols (BBA) on the tropical African climate in an ocean-atmosphere-aerosols coupled climate model."

_EGUsphere, 2024_

## Author Comment (AC1)

OVERALL COMMENTS:

Marc et al. used an coupled climate model to investigate the impact of biomass burning aerosols (BBA) on (1) cloud fraction, (2) ocean temperature and lower tropospheric stability, (3) AOD and surface solar radiation, (4) precipitation. While I appreciate the value of a coupled model, the quantitative results are not reliable and the attributions to processes are unconvincing given the limitations provided below. Besides, the presentation quality of the paper is very low. The use of terminology is inconsistent. Figures need panel titles, labels, and so on to improve clarity.

**First of all, we would like to thank the reviewer for his numerous relevant comments. In the new version, we have taken them into account to hopefully improve and clarify the text of the article. In particular, we now clearly discussed the limitations of our study, especially with respect to the possible impact of low-level cloud cover biases and the absence of certain processes (ageing and the indirect radiative effect). We also updated the text and some figures in order to better discuss the main causes/attributions of the identified impacts of BBA radiative effects on low-level clouds and precipitation. For that purpose and in order to better quantify the role of the ocean-atmosphere coupling in the response of several variables, we carried out a new set of simulations (in SST forced mode) to investigate the contribution of the radiative effects of smoke aerosols relative to the role of the influence of SST. Finally, we now highlight in the introduction and conclusion some of the original features offered by the use of coupled modelling, in particular concerning the impact of the BBA radiative forcing on the ocean temperature and on the main surface currents.**

MAJOR COMMENTS:

Like many other global models, the model used in this study has significant low bias in low cloud fraction in the area of interest (Roehrig et al. 2020). The horizontal resolution is lower than many other GCMs/ESMs nowadays. How do these limitations affect all the quantitative results/arguments related to low clouds ?

**We agree that the representation of low-level clouds over this region in climate models may limits our conclusions about the impact of BBA radiative effect. To our knowledge, the low cloud bias at the eastern edges of the subtropical oceans still challenges most global models whatever their horizontal resolution (e.g., CMIP6, e.g. Figure S3 of Crnivec et al., 2023 from https://agupubs.onlinelibrary.wiley.com/doi/10.1029/2022JD038437). Based on the reviewer remark's, we now further discuss this specific limitation in the text and conclusion. Additional explanation was also added in Parts 3.1 and 3.3. As stated in the paper, one of the main limitation is that the bias in low-level clouds leads to underestimate the reflectivity of the surface beneath the BBA during the transport over the ocean. This will limit the reflection of solar radiation and therefore decrease the absorption of radiation by BBA and radiative heating. We now highlight this important point based on the study by Feng et al. (2015) (see response below), showing that for AOD of ~1 and SSA of ~0.90 at 550 nm (typical of the values observed in this region), an increase in cloud optical depth induces a decrease in the SW direct radiative effect for smoke aerosols above clouds at TOA. This indicates the enhancement of solar absorption by BBA as cloud reflectivity increases, which could contribute to an increase in radiative heating by smoke aerosols. This specific point is now indicated and could explain the fact that the simulated heating rates in CNRM-CM are somewhat lower than the values estimated over this region. Since this radiative heating partly controls the response of the low-level clouds by limiting the supply of dry air within the marine boundary layer, it is likely that the response of low clouds is underestimated in our simulations. At the same time, it is also possible that this underestimation of low clouds and therefore of radiative heating influences the response of atmospheric dynamics over the Western Africa, with an underestimated response of precipitation (and notably the drying effect). We finally now indicated in the conclusion that it would be relevant to carry out a multi-model experiment to better quantify these limitations, but also to better assess the robustness of our results. All these limitations are clearly set out in the conclusion.**

**Crnivec. N., Cesana, G. and Pincus R., Evaluating the representation of tropical stratocumulus and shallow cumulus clouds as well as their radiative effects in CMIP6 models using satellite observations, Journal of Geophysical Research: Atmospheres 128 (23), e2022JD038437, 2023.**

**Concerning the horizontal resolution of the model used in this work (~1.4°), it is found within the average of current CMIP6 models (https://link.springer.com/article/10.1007/s41748-020-00157-7/tables/1). It does exist CMIP6 models with higher horizontal resolutions (e.g., 50 km) but to our knowledge, no study has demonstrated any added value of such higher resolution to the representation of low-level clouds.**

The model assumes the aerosols to be externally mixed. This is unrealistic for aerosols undergoing long distance transportation. In particular, there is no second indirect effect. Then it is not surprising that much of the impacts of BBA on several aspects of the model came from direct and semi-direct effects. How do these limitations affect the results ?

**We agree that the fact that aerosols are treated as externally mixed represents an important hypothesis as it does not consider explicitly the ageing and possible coating processes of the BBA during the transport. However, in the context of this work, the lack of representation of this specific process is not necessarily a limitation. To this end, we verified first that the SSA simulated by the CNRM-CM model was consistent with recent observations over this region (see Figure 1b), ensuring that the BBA simulated over southern Atlantic are highly absorbing. This means that the radiative heating associated with BBA is also well considered (as reported in Figure 2b), which is an essential requirement for studying the direct and semi-direct radiative effects of BBA. Thus, even if we recognise that this important process is absent in the current version of the model, the main feedbacks; i.e. solar absorption and radiative heating due to BBA, have been evaluated and are well taken into account (we have also now clearly indicated in the part 3.1 that the underestimation of the low-cloud cover could lead to an underestimation of the heating rate, see previous question). Furthermore, it should be noted that this simplified representation of the aerosol mixing state in CNRM-CM allows also to run longer climate simulations which is needed in the coupled mode. All these points are now emphasised in the part 2.2.**

**In addition, the second indirect effect is indeed not represented in the CNRM-CM model, which is the case for the majority of current global climate models. The very complex processes involved in the second indirect effect are generally accessible in very high spatial (~km) resolution models that explicitly represent convection and the interactions between hydrophilic aerosols and clouds. We agree that we have effectively not emphasised enough this limitation to represent the possible impacts of BBA on the cloud microphysics and precipitation. This was partly based on various recent results obtained over this region, which generally indicate that the effects of BBA solar absorption outweigh the interactions with microphysics. As an example, Che et al. (2021) have shown that the absorption effect of BBA is the most significant on clouds and radiation over the SEA using the UK Earth System Model ($1.875° \times 1.25°$ horizontal resolution), which includes the first and second indirect aerosol effects (Mulcahy et al. 2020). They have shown that the liquid water path over the SEA is significantly enhanced, mainly due to the solar absorption of the BBA, especially when located above the stratocumulus clouds. Using the WRF-Chem-CAM regional model with large eddy simulations, Diamond et al. (2022) found a significant increase in cloud cover during a given event when all smoke effects are included, mainly driven by large-scale thermodynamical and dynamical semi-direct effects. Finally, and at the climate scale, Solmon et al. (2021) showed that the "microphysical" radiative effect is relatively weak compared to the direct/semi-direct effects on the cloud and precipitation responses (although the authors note that the contribution of the indirect effects should be taken with caution due to a rather simplified representation in climate models). Again, we recognize that this limitation was not enough detailed and these important points are now all included in Part 2.2.**

**Che et al.,: Cloud adjustments dominate the overall negative aerosol radiative effects of biomass burning aerosols in UKESM1 climate model simulations over the south-eastern Atlantic, Atmos. Chem. Phys., 21, 17–33, 2021**

**Mulcahy et al.,: Description and evaluation of aerosol in UKESM1 and HadGEM3-GC3.1 CMIP6 historical simulations, 13, 6383-6423, https://doi.org/10.5194/gmd-13-6383-2020, 2020.**

**Diamond, M. S., et al.,: Cloud adjustments from large-scale smoke–circulation interactions strongly modulate the southeastern Atlantic stratocumulus-to-cumulus transition, Atmos. Chem. Phys., 22, 12113–12151, https://doi.org/10.5194/acp-22-12113-2022, 2022.**

**Solmon, F., et al., Modulation of West African Monsson Precipitation by Central and Southern African Biomass Aerosol Emissions, npj Climate and Atmospheric Science, 4:54, 2021.**

MINOR COMMENTS:

Section 2.3: What emission inventory was used?

**The emission inventory GFED4 is used in this study. This important point is now indicated in the text including the following reference (van Marle et al., 2017).**

**van Marle, et al.: Historic global biomass burning emissions for CMIP6 (BB4CMIP) based on merging satellite observations with proxies and fire models (1750–2015), Geosci. Model Dev., 10, 3329–3357, https://doi.org/10.5194/gmd-10-3329-2017, 2017.**

L169: "which could be due to ...": The authors listed so many possible causes that the sentence lost its value. Please narrow down the cause.

**This is right and we modified the text as follows. The changes in AOD over the Arabian Peninsula is attributed to a small decrease in the precipitation (Fig. 8a) and not to changes in the surface wind (Fig. A4). The new sentence is the following : «Some negative AOD anomalies appear locally especially over the Arabian Peninsula, which are mainly due to negative feedbacks of the BBA radiative forcing on precipitation that favouring wet deposition and decreased emissions of mineral dust particles over this region.»**

L193: "This difference is possibly attributed to ...": Can this hypothesis be tested using some offline radiative transfer model?

**This is a very good comment. To answer it, we have not performed specific offline radiative transfer simulations but we take advantage of the study of Feng and Christopher (2015) to underline that the bias in cloud albedo could explain the slight underestimate of BBA radiative heating in the CNRM-CM model. In this sense, we used Table 2 of Feng and Christopher (2015), who investigated the radiative forcing of BBA over this region by performing sensitivity tests with a radiative transfer code. This study shows that for an AOD of ~1 and SSA of ~0.90 at 550 nm (typical of the values observed in our case study, Figure 1a,b), an increase in cloud optical depth from (from ~6-8 to ~8-12) induces a decrease (by about ~-10 W.m$^{-2}$) in the SW direct radiative effect for aerosols above clouds at TOA. This decrease indicates the BBA solar radiation due to higher cloud albedo that contribute to the intensification of solar radiative heating by smoke aerosols. This point is now clearly indicated in the text: «This difference is possibly attributed to the under-estimation of low-level clouds over southeast Atlantic in the CNRM-CM model (Brient et al., 2019), limiting the reflection of solar radiation by clouds and hence solar absorption by BBA plumes. _Indeed, Feng and Christopher (2015) showed that an increase in the cloud optical depth by about ~2-4 leads to a decrease (of about ~-10 W.m$^{-2}$) in the SW direct radiative effect for smoke aerosols (characterized by AOD and SSA of ~1 and ~0.9 at 550 nm, respectively) above clouds at TOA. This reflects the additional solar absorption by the BBA due to higher cloud reflectivity that could then contribute to the enhancement of solar radiative heating by smoke aerosols._»**

**Feng, N., and S. A. Christopher (2015), Measurement-based estimates of direct radiative effects of absorbing aerosols above clouds, J. Geophys. Res. Atmos., 120, 6908–6921, doi:10.1002/2015JD023252.**

L210: The mechnism here is over simplified. The LTS increases in a broad area. But the regions with increases in Figures 3a, 3b, and 3c all show two stripes. In other words, an increase in LTS does not guarantee an increase in low cloud fraction or LWP. Please explain.

**The different processes identified over the ocean were indeed unclear and not enough detailed and discussed in the text. For the low cloud fraction, the LTS increase (due to the SST decrease and the atmospheric radiative heating) seems to be the main process explaining the increase of LCF, with a good regional correlation between the two parameters. The highest impact simulated over the SEA, compared to the Gulf of Guinea, corresponds well to the more pronounced LTS changes on this region. However and as remarked by the reviewer, the impact on LWP is not directly explained by the LTS changes. Over the SEA, as most of clouds are low-level clouds, the correlation still exist and the LWP increase due to lower intrusions of dry air within the marine boundary layer. In opposite, this effect is different in the Gulf of Guinea while the LTS is also increased. This is due to a compensation between an increase of the cloud liquid water between the surface and 850 hPa and a decrease at ~600 hPa (see new Figure A5). In parallel, the LWP negative anomaly along the West African coast is not related to the change in the LTS (no significant signal over this zone, see Figure 4c) but to the impacts of the BBA radiative effect on the atmospheric dynamics (southward shift of the rainbelt), as explained in part 3.3. These points are not discussed in the part 3.2.1 «Unlike the LCF, the effect of the LTS on the LWP is not as direct, as for example in the Gulf of Guinea region, where the effect is smaller even though the LTS also increases (Figure 4c). In this region, this is due to the compensation between an increase in cloud liquid water content between the surface and ~850 hPa and a decrease at 600 hPa (Figure A5).»**

[Figure]

**New Figure A5. Vertical profiles of the Mass fraction of Cloud Liquid Water (in kg kg$^{-1}$) anomaly for the longitudinal transect averaged between 2°S and 2°N.**

L230: "This could be due to": Please elaborate. How similar/different are the surface radiations in Solmon 2021 and current work ?

**This is a good remark. We compared the simulated surface direct forcing in the RegCM and CNRM-CM models. It is more intense in the RegCM model with in particular a much larger extension to the west and over the Gulf of Guinea (see Fig. 7a in Mallet et al. 2020). This should explained why the impact of BBA on SST calculated in the RegCM-SOM is higher in Solmon et al. (2021). This important point is now clearly indicated in the text (line 266) : «This could be due to the slab ocean model vs. 3D oceanic model and to the strongest dimming simulated over the ocean in the RegCM model due to higher smoke optical depth and low cloud response.».**

L233: "local heating": What does "local" mean here?

**The term « local » would express that the atmospheric radiative heating was occuring within the smoke aerosol plume. This is now detailed in the text: «Both the solar heating at ~600-800 hPa and the SST decrease contribute to increasing the LTS (Figure 4c)».**

L252: I assume there are many differences between the models used in these studies. Please explain why the treatment of the ocean and ocean-atmosphere coupling is believed to be the most important factor.

**This point was effectively not enough detailed and we have now provided more informations on the different models used. Concerning the SST-forced simulations (Mallet et al. 2020 and Allen et al 2019), we indicated that the BBA surface radiative forcing and solar heating are found to be more important in these two studies over the Gulf of Guinea compared to the CNRM-CM simulations. This could also impact the LCF in addition to the Ocean-Atmosphere coupling over this region. In parallel, we used the new forced simulations to show that the coupling could explain part of the difference over the Gulf of Guinea. In that sense, we now moderate the conclusions in the text (lines 296-298): «This is found to be consistent with the results indicated in Fig. A6, which shows that the impact on SST clearly affects the low cloud fraction (increase up to + 5%) over part of the GG. In parallel, another source (other than the O-A coupling) explaining these differences in the response of low-level clouds may be the BBA surface radiative forcing, which is found to be greater over the GG in the SST-forced models, associated with higher solar heating ». At the end of this paragraph, we now indicate : « Although it is difficult to draw a final conclusion, especially due to differences between the models, the inclusion of the O-A coupling seems to lead to an increase of low-level clouds over the GG in contrast to the SST-forced simulations.»**

L260: "However, ...": Please be quantitative.

**The sentence has been updated : « However, the amplitude of the low-cloud fraction response is found to be stronger in the coupled model (increase of about ~10%) over the SEA than in the RegCM-SOM (increase of ~3-5%) model, with an impact of up to 15°W in the CNRM-CM simulations.»**

L267: "where the negative anomaly may exceed that identified between the surface and 20 m.": Would you please provide an explanation for this feature?

**It's very difficult to answer this interesting question without a specific study of the ocean response, which we plan to do in the future. In fact, the aim here was to show the possible first-order effects of the smoke radiative forcing on certain ocean variables. We are now going to study the modification of ocean dynamics, density and salinity, especially for the different ocean layers where the temperature response to the BBA forcing is relatively important.**

L286: "Over Central Africa, ...": Would you please provide an explanation for this feature?

**This specific results obtained over the Central Africa is detailled at the lines 337-339 : «In parallel and over Central Africa, the decrease in surface air temperature over the continent (Figure 4a), in addition to diabatic heating, leads to the stratification of the lower troposphere and limits convection (Figure 6b), which helps to maintain low cloudiness.»**

L288: "due to the coupling between the ocean and the atmosphere being taken into account": Please elaborate.

**This specific point indeed deserved more details and better quantification. By using new SST forced-mode CNRM-CM simulations (see new Table 1), we are able to show that the effect of coupling is relatively limited for the low-level cloud response over the coastal region of Angola and Gabon, in contrast to the oceanic region where the effect of SST is important (see new Figure A6). Hence, these new simulations show that the increase in low-level cloudiness over the coastal regions is rather related to the increase in moisture advection than to the coupling between the ocean and the atmosphere. In that sense, we have:**

**→ detailed the new CNRM-CM simulations in the part 2.3 : « To disentangle the direct effect of BBA on the atmosphere and the feedbacks resulting from SST changes, three additional SST forced simulations have been performed. The first two experiments, (ATM-ref and ATM-BBA-SST, see Table 1) are twin experiments of the coupled experiments, without and with BBA radiative effects where the SST forcing is taken form the respective coupled experiments (Table 1). The third experiment (ATM-BBA-ref) combines the SST from the ATM-ref experiment and the BBA forcing. To summarize, the difference ATM-BBA-ref minus ATM-ref indicates the impact of BBA when SST are fixed, whereas the difference ATM-BBA-SST minus ATM-BBA-ref indicates the additional impact due to the SST change. In the following, the analyses of the simulations are mainly focused on the coupled configuration, and the additional forced simulations are used to analyse the contribution between the direct radiative forcing of the BBAs and the effects due to the change in SST.»**

|  | BBA | Atlantic SST |
|---|---|---|
| CPL-ref | No | coupled |
| CPL-BBA | Yes | coupled |
| ATM-ref | No | SST from CPL-ref |
| ATM-BBA-ref | Yes | SST from CPL-ref |
| ATM-BBA-SST | Yes | SST from CPL-BBA |

**→ modified the conclusions for the low-level clouds over the continent: «As shown in Figure A6, the response of the low-level clouds simulated in the coastal areas of Gabon and Angola is not sensitive to the coupling between the ocean and the atmosphere  and is probably more related to the increase in moisture advection over this region (Figure A6), which contributes to enhance the low cloud fraction. »**

L321: "the BBA radiative effects and solar heating": Isn't solar heating part of the radiative effect?

**This is right and now changed in the text by removing « and solar heating ».**

L355: "Indeed and as mentioned previously, ...": There are also regions with positive humidity anomaly but decreased precipitation. Please explain.

We agree that some regions (such as the Congo) show a decrease in rainfall while relative humidity is increasing. However, the anomaly obtained over this region is not significant in the sense of the Wilk's test (Figure 8a), which is why we have preferred not to discuss the results obtained for the precipitation over this specific region.

L362: The stongest positive precipitation anomaly occurred along the coast of Gabon and extended to the west. Why analyzing the moisture tendency anomaly for Angolan coast ?

**As indicated above, we have decided not to discuss the positive changes over Gabon too much, since this region is not statistically significant and therefore the "physical" signal is not necessarily realistic and due to the smoke radiative forcing. We have therefore decided to concentrate our analyses on Angola, where the signal is less important but statistically robust.**

L385: "with a more pronounced effect": Would you please provide an explanation for this feature?

**This particular point was effectively not detailed enough. This response is mainly due to the fact that the SST anomaly is largest off the Atlantic ocean, with a maximum (ΔSST of -0.5 K) around 15°W (Figure 10b). This then affects the surface air temperature up to the coast of Brazil (Figure 10a) and explains why the effect on the cloud fraction over this region is important. In parallel, the simulations indicate also a slight positive anomaly of the relative humidity along the coast of Brazil at 1000 and 925 hPa (see the new Figure A10 in the new version) that contributes also to increase the low-level clouds over this region. In that sense, we have modified and tempered the conclusions by modifying the following sentence : «The results  indicate the existence of a gradient in the low-cloud response across the ocean basin, with an  important effect between 15°W and the coasts of Brazil compared to the response simulated on SEA. This is partly due to the SST anomaly which is largest off the Atlantic with a maximum (-0.5 K) around 15°W (Figure 10b). This anomaly then affects the surface air temperature up to the coast of Brazil (Figure 10a). At the same time, the simulations show a slight positive anomaly in the relative humidity over a large part of this region (Figure A10 in the Appendix), which also contributes to increase the low cloud fraction.»**

L386: "Contrary to JJA, ...": The statement in this sentnence is inconsistent with Figure 9a.

**This is right and now changed in the text by: «The Figure 9a also shows that the low cloud fraction over the coastal areas of Gabon is less influenced by biomass burning emissions during the SON season compared to the JJA.»**

L393: The attribution of the strong signal outside of the Brazilian coast to the mission changes in Amazon is not convincing. Please elaborate.

**This is a good remark but it is difficult to answer without performing additional simulations without Amazonian biomass burning emissions. However, Figure A1 clearly shows that the largest AODs simulated over the Amazon do not coincide with the regions where the impact on cloud cover is significant (particularly over the northern Brazil and Guyana). It is therefore likely that the simulated anomalies of cloud properties in these regions are associated with a change in atmospheric dynamics that may alter the advection of moisture over this region. As indicated in the text, this point will be investigated in a specific study by modifying the biomass-burning emissions. However, we have now modified and tempered the conclusions in this sentence: «As shown in Figure A1, the largest AOD due to smoke aerosols in Amazonia are simulated further south of the region where effects on cloud properties are significant. It is therefore possible that the main effect of Amazonian emissions are not related to radiative processes but rather to a change in atmospheric dynamics affecting moisture advection over these regions.»**

TECHNICAL ISSUES:

Figure 2b: There is no need for the degree symbol with K.

**This is now changed.**

Figure 6a: What is "medium-cloud"?

**The « medium-cloud » refers to « mid-level cloud » which correspond to clouds located between the 785-450 hPa atmospheric layer. This is now detailed in the text.**

Figures 6b and 6c: Assuming the gray shading is terrain, why is it different in 6b and 6c, both averaged between 15 deg E and 25 deg E and shown for about the same latitude range?

**This difference is due to the horizontal resolution of the two variables (vertical velocity and cloud fraction) which are not exactly similar in the CNRM-CM model outputs.**

Figure 7: Which one is 925 and which one is 850 hPa?

**Sorry for this mistake, this is now indicated in the legend of the Figure 7.**

Figure A3: Are the panels in the left column for simulations with or without BBA emissions ?

**The left column corresponds to the CNRM-CM simulations including BBA emissions. This is now indicated in the legend.**

L346: Unit seems wrong.

**This is right and now changed.**

Figure 8: Units not fully visible.

**This is now modified.**

L380: What does "As for" here?

**This is now removed from the sentence.**

L420: What does "In parallel and as for the JJA season" mean here?

**This was effectively not clear and we have now removed «and as for the JJA season».**

L422: "-0.3": Unit missing

**This is now modified.**

The authors used "beyond" some longitudes in many places. Do they all mean "to the west of"? Please clarify. Also, in L235: "below the equator": Does it mean "to the south of the equator"?

**Yes, effectively, the term « beyond » has been used to indicate « to the west of ». This is now changed in the text. In parallel, we have also modified « below the equator ».**

Need to explain what is RegCM-SOM model and what are RegCM simulations. Are "RegCM-SOM" and "RegCM" the same thing?

**This was effectively a mistake. References to the RegCM model in the study always refer to the RegCM atmospheric model coupled to the Slab Ocean Model (SOM). There are no references to the RegCM atmospheric model alone. This has now been changed in the text by retaining only the RegCM-SOM model.**

---

## Author Comment (AC2)

**General Comments:**

This study evaluates the coupled effects of biomass-burning aerosols from southern Africa on the regional climate, simulated by the CNRM-CM model over the period from 1990-2014. The large mass of emissions, combined with their steady presence for several months, exert strong effects on clouds, radiation, and the sea surface temperature. By comparing with a model run without these smoke emissions, the study shows that there are large and distinct effects in JJA (which is the majority of the annual burning period) and SON. These effects drive large local uncertainty in radiative balance, and this work provides an important analysis on when, where, and how BBA effects manifest. The paper is well-written and makes a convincing argument especially for the value of a dynamical ocean model in capturing regional trends outside of the main burning season. There are some points of clarification and background that do not detract from the overall work, and I recommend publication after minor corrections.

**First of all, we would like to thank the reviewer for his relevant comments. In the new version, we have taken them into account thereby improving and clarifying the text and figures of the article.**

**Specific Comments**

Introduction:

As this study is a model application, studying the effects of toggling BBA emissions on and off, it necessarily can't avoid inherent model biases. The authors show that the model represents smoke SSA well, but I would like to see some comment on model performance for other properties central to the study, where available, such as other smoke attributes or placement, cloud properties, or winds.

**This is indeed an important point. We now provide information on the main features of the CNRM-CM model in terms of BBA, cloud properties and wind fields. Regarding BBA, the simulations show maxima (AOD ~ 0.7 at 550 nm) over Congo and Angola with a plume covering the whole SEA during the JJA season with a strong decrease in AOD at ~15°W (Figure 1a). This regional pattern is in relatively good agreement with spatial AOD satellite inversions or reanalysis products, as shown in Mallet et al. 2020. This specific point is now clearly indicated in the part 3.1. With regard to clouds, we now point out that the CNRM-CM model suffers to represent low-level clouds over this region. Such a bias occurs in many state-of-the-art model (e.g., CMIP6, e.g. Figure S3 of Crnivec et al., 2023 from https://agupubs.onlinelibrary.wiley.com/doi/10.1029/2022JD038437). This point has been added and the possible implications are discussed in the part 3.1. In particular, we indicate that this limits the reflection of solar radiation and therefore decrease the absorption of radiation by BBA and radiative heating. We now highlight this important point based on the study by Feng et al. (2015), showing that for AOD of ~1 and SSA of ~0.90 at 550 nm (typical of the values observed in this region), an increase in cloud optical depth induces a decrease in the SW direct radiative effect for smoke aerosols above clouds at TOA. This indicates the enhancement of solar absorption by BBA as cloud reflectivity increases, which could contribute to an increase in radiative heating by smoke aerosols. This point is now indicated and could explain the fact that the simulated heating rates in CNRM-CM are somewhat lower than the values estimated over this region. Finally, we integrated the mean wind fields at 950 and 850 hPa over the JJA season in Figure 7. These new figures clearly show the southwesterly flow over the tropical Africa, which is characteristic of the region and responsible for the development of the West African monsoon in JJA. This point is now indicated in the part 3.3.**

**Crnivec. N., Cesana, G. and Pincus R., Evaluating the representation of tropical stratocumulus and shallow cumulus clouds as well as their radiative effects in CMIP6 models using satellite observations, Journal of Geophysical Research: Atmospheres 128 (23), e2022JD038437, 2023.**

Feng, N., and S. A. Christopher (2015), Measurement-based estimates of direct radiative effects of absorbing aerosols above clouds, J. Geophys. Res. Atmos., 120, 6908–6921, doi:10.1002/2015JD023252.

Lines 63-64: I understand that "few" is relative, but there are multiple recent studies overall analyzing the impact of African BBA on clouds, dynamics, and precipitation in the region. There have been several modeling studies addressing aspects of this question in the last several years with various methods, such as following the field campaigns AEROCLO-SA, ORACLES, CLARIFY, or LASIC. These may have important differences with this work, but they remain studies of this region on these topics. For example: Lu et al 2018, Gordon et al 2018, Diamond et al 2022, Perez et al 2023.

**We agree with this comment and we have now modified the sentence in the introduction as follows : « In parallel to the interactions between desert dust aerosols and the hydrological cycle over Tropical Africa (Solmon, 2008, 2012; Balkanski et al., 2021), different studies have addressed the impact of BBA plumes emitted over central Africa on cloud properties, atmospheric dynamics and precipitation in the tropics (Lu et al., 2018; Gordon et al., 2018; Diamond et al., 2022; Chaboureau et al., 2022 and Baró Pérez et al., 2024). Recently, Solmon et al. (2021) and Ajoku et al. (2019)...»**

Methods:

Please add some physical description of the different size modes, such as the central diameter of each size bin. Aerosol optical and microphysical processes depend heavily on size ranges and this will give better context to other studies comparing to this work with different size schemes or parameters.

**We now detail this specific point in the part 3.1. The central effective radius for natural desert dust (0.1, 0.83 and 5.8 µm) and sea salt (central effective radius of 0.15, 1.9 and 19.1 µm) are provided. We also mentioned the Rémy et al. (2022) reference which indicates the parameters of the size distribution for other aerosol species (organic matter, black carbon, sulfates, nitrate fine/coarse and ammonium) used in TACTIC.»**

Rémy, S., Kipling, Z., Huijnen, V., Flemming, J., Nabat, P., Michou, M., Ades, M., Engelen, R., and Peuch, V.-H.: Description and evaluation of the tropospheric aerosol scheme in the Integrated Forecasting System (IFS-AER, cycle 47R1) of ECMWF, Geosci. Model Dev., 15, 4881–4912, https://doi.org/10.5194/gmd-15-4881-2022, 2022.

Line 132 and 142: Are nitrates and ammonium considered hydrophobic or hydrophilic, or something else?

**Ammonium-nitrates particles are considered as hydrophylic. This is now indicated in the text and the reference of Druge et al. (2019) describing this aerosol species has been added.**

Since precipitation changes are one of the focus topics of this work, I would like to see some mention of the impact of the missing second indirect effect as a standing uncertainty that could possibly modulate these results.

**You are fully right that the second indirect effect is not represented in the CNRM-CM model. Note that, to our knowledge, this is the case in the majority of global climate models. The very complex processes involved in the second indirect effect are generally accessible in very high spatial (~km) resolution models that explicitly represent convection and the interactions between hydrophilic aerosols and clouds. We agree that the implication of missing this process deserve further discussion, now added in Part 2.2. Several recent studies emphasize that the effects of BBA solar absorption outweigh the interactions with microphysics. As an exemple, Che et al. (2021) showed that the absorption effect of BBA is the most significant on clouds and radiation over the SEA using the UK Earth System Model, which includes the first and second indirect aerosol effects (Mulcahy et al. 2020). They showed that the liquid water path over the SEA is significantly enhanced, mainly due to the solar absorption of the BBA, especially when located above the stratocumulus clouds. Using the WRF-Chem-CAM regional model with large-eddy simulations,**

**Diamond et al. (2022) also indicated a significant increase in cloud cover for a given event when all smoke effects are included, mainly driven by the large-scale thermodynamic and dynamic semi-direct effects. Finally, at the climate scale, Solmon et al. (2021) showed that the "microphysical" radiative effect is relatively weak compared to the direct/semi-direct effects on the cloud and precipitation response (although the authors note that the contribution of the indirect effects should be taken with caution due to a rather simplified representation in climate models).**

**Che et al.,: Cloud adjustments dominate the overall negative aerosol radiative effects of biomass burning aerosols in UKESM1 climate model simulations over the south-eastern Atlantic, Atmos. Chem. Phys., 21, 17–33, 2021**

**Mulcahy et al.,: Description and evaluation of aerosol in UKESM1 and HadGEM3-GC3.1 CMIP6 historical simulations, 13, 6383-6423, https://doi.org/10.5194/gmd-13-6383-2020, 2020.**

**Diamond, M. S., et al.,: Cloud adjustments from large-scale smoke–circulation interactions strongly modulate the southeastern Atlantic stratocumulus-to-cumulus transition, Atmos. Chem. Phys., 22, 12113–12151, https://doi.org/10.5194/acp-22-12113-2022, 2022.**

**Solmon, F., et al., Modulation of West African Monsson Precipitation by Central and Southern African Biomass Aerosol Emissions, npj Climate and Atmospheric Science, 4:54, 2021.**

Line 149: Add a comment that defines the term "anomaly" used throughout the paper as in reference to the difference between these models, and exactly how it is being calculated.

**We have now clarified this point by modifying the following sentence : « In the results presented thereafter, all the anomalies analysed for different variables correspond to the differences between the CNRM-CM simulations with and without the biomass-burning emissions. In addition, the statistical test applied is the Wilks test (Wilks, 2006, 2016) to ensure the robustness of the results. »**

Results:

193-194: Is the modeled SSA being 0.03-0.08 higher than observations playing a part in this heating differential ?

**This is an excellent point that could contribute to the underestimate of solar heating rate due to BBA. This is now included in the text «This may be due to a slight over-estimation of the BBA SSA during the plume transport over the SEA».**

**Technical corrections**

Figure formatting:

- Figure titles have "Anm" in the title but not defined.
- Several figure axes are labeled with the word 'Presion', which I believe should be 'Pressure'
- The dashed grid lines for lat/lon should be labeled in most or all figures
- Since every model being used here is CPL_ndg, it isn't necessary in figure titles since it doesn't differentiate anything.

**All the figures have been modified following the different remarks.**

"Positive feedback" and "negative feedback" are used in multiple places when the context suggests the authors intend to mean 'Positive/negative **effect**' instead. The usage of feedback implies to me that the effect is self-reinforcing or self-destroying via some mechanism, rather than simply reporting an increase or decrease of some quantity. (examples at least at lines 5, 18, 415, 441, 451, )

**We agree with this remark and the term « feedback » has been changed by « effect » in the text.**

Line 48: "indicate" should be "indicates"

**This is now changed.**

Line 66: Should read "From the methodological…"

**This is now modified.**

Line 95: Confusing sentence structure about what is causative and what is impacted- consider rewriting as "The overall effect on the solar surface radiative budget by both the BBA direct effect and changes in tropical clouds is also discussed."

**This is now changed in the text.**

Line 106: Is the second mention of "carbon cycle" redundant ?

**This is right and now removed in the new version.**

Line 108: missing right parenthesis )

**Now changed in the text.**

Line 128: ambiguous usage of "supposed" - do you mean "assumed"? Or "intended"?

**This term is effectively not adapted. We have now used « assumed » in the new version.**

Line 128: "Externally mixed" does not refer to aerosol particles being separated by sources, but by species. I.e., a single particle is composed of a single species.

**This is right and now changed.**

Line 152: Should read 26N, not 26S

**This is now modified.**

Line 159: If this applies to every simulation used here, I don't see the need to specify a new acronym and put it in figure titles, as that would lead me to expect an un-coupled or un-nudged configuration to come up.

**This is right. We have now removed the acronym in the text and for all figures.**

Line 162: Replace "thereafter" with "hereafter" or "below"

**This is now changed in the text.**

Line 165: Should read "…anomaly for JJA shows…" without the 'the'

**This is now modified.**

224: Change to "…anomaly is low west of 5°W…" rather than 'above'

**This is now changed in the text.**

225: Specify what LW radiation means here - it seems to mean downwelling LW emissions from clouds, but is not clear the source.

**This is effectively right and mean the downwelling emissions from clouds. This point has been detailled in the text.**

272: 'Atlantic coast' should read 'African coast'

**This is now changed in the text.**

288: 'low cloud response' is ambiguous. Does it refer to the response of low clouds? Or the relatively *weak* cloud response ?

**This was effectively not clear and refers to the response of low-level clouds. This is now changed in the text.**

329: Should read '...the ocean modulates the BBA…" not 'modulate'

**This is now changed.**

337: The wording is confusing with "on the other hand", since both results come from CNRM-CM. Perhaps change to "On the other hand, there is a moderate positive impact over northern Angola in CNRM-CM simulations."

**This sentence is now modified in the text.**

350: What do you mean 'more important'?

**This is effectively not enough clear and we have now modified this sentence : «...in particular the drying over the coastal regions of Liberia, Sierra Leone and Guinea which is more important in this study (-1.5 mm by day) compared to the RegCM simulations (-0.6 mm by day).»**

428: What does precipitation 'by day' mean? Use either 'per day'/'daily', or 'during the daytime' depending on what you are saying.

**This was effectively wrong and the unit is mm.day$^{-1}$. This is now changed.**

429: change 'on' to 'in'

**This is changed.**

---

## Referee Report (RR1)

**Referee report**

General comments:

This is a second review of 'Impact of Biomass Burning Aerosols (BBA) on the tropical African climate in an ocean-atmosphere-aerosols coupled climate model' after authors revised in response to the first round of referee comments, and comments largely focus on the new or altered content. The authors' additional comments and especially new simulations provided addition context to help disentangle effects of the dynamic SST and the aerosol radiative effects on cloud properties and ERFs, improving the quality of the analyses against the first version. I recommend acceptance after minor revisions.

Specific comments:

Minor:

The simulations to separate the impacts from BBA radiative dynamical effects vs. surface SST effects on cloud properties are very useful. As these are the dominant proposed mechanisms at play here, the differentiated models for BBA and SST effects should be mentioned in more places as appropriate through the work to justify claims of whether one effect or the other is dominant. For example:

- 346: Doesn't figure A6a) show that the BBA effect is strongly tied to the increased LCF over the Gabon/Angola coast? Is the increased moisture advection tied to BBA radiative impacts then?

- 475: The BBA effect being weaker isn't simply 'more likely', it is explicitly backed up by figure A6. I recommend mild reorganization to move the claim and the evidence (~line 487) closer together.

513: I read this sentence as claiming that the SST decrease is the dominant effect on changing cloud patterns, but figure A6 seems to show the opposite-- the BBA effect is at least as strong in JJA, if not stronger, over a wider ocean area and is the dominant influence near Gabon.

Technical corrections:

23: feedback should be 'effect' on precipitation

142: missing left parenthesis for Druge 2019

149: Biomass Burning shouldn't be capitalized

181: should read "the present configuration allows *us* to focus on solely the…" with no comma

206: favoring should be 'favor'

276: 'strongest' should probably be 'stronger'

304: define 'GG' and 'O-A' coupling

314: 'with an impact of up to 15W' as a phrase doesn't make sense. '…with an impact of up to [magnitude] at 15W', or perhaps 'with an impact out to 15W' as possible alterations.

Section 3.3: should read 'dynamics', not 'dynamic'

Figure A8 shouldn't have underscores in the title

400: Clarify the direction of the model bias with '…biased towards underestimating low-level cloud cover', as long as that is what's intended

420: figure reference should be A7

461: should read 'do not allow **us** to disentangle…'

474: Figure reference should probably be A9, not A8

487: remove 'the' before 'Figure A6'.

489: I believe the reference to figure A7 should probably be A6 for SON cloud anomalies

498: Clarify to say there are no major changes over **the African continent**. South America shows significant differences in precipitation.

559: either write 'using different GCMs' or, less favored because it's redundant with the acronym (global circulation model models), 'using different GCM models'

508: indicate should be 'indicates'

513: Suggest a rephrase to clarify and organize, the original sentence is confusing about what is cause and what is effect(s). One suggestion: This positive impact is found to be mainly due to the SST decrease, which is in response to the surface BBA radiative forcing ($\sim$-5 to -15 W.m$-2$) and the cloud changes associated with lower tropospheric heating. These both contribute to (i) increasing the LTS and (ii) to limiting the intrusion of dry air at the cloud top.

---

## Author Response (AR2)

**Dear Editor,**

**In the new version, we have taken into account all the points listed below.**

The manuscript is in a good shape but needs some technical corrections found by Reviewer 1. Also, please do the following technical corrections:

- Change from "Mallet Marc" to "Marc Mallet" in the author list
*This is changed.*

- line 41: "thought be" => "thought to be"
*This is modified.*

- line 154: use the super script for the degree symbols in "1.875o x 1.25o"
*This is now changed.*

- line 157: "Large Eddy" => "large eddy"
*This is changed.*

- line 228: "decrease (of about -10 W.m^-2)" => "decrease (of about 10 W m^-2)" I found that two mistakes in English and style throughout the text and this is a good example. Use magnitude after "decrease of", "reduction of", "decreased by", "reduced by", and "the order of". For these expressions only magnitude is used, e.g., decrease of 10. Your don't have to use "+" with expressions like "increase of". Remove "." between unit symbols and use a space, e.g., m s^-1. Please fix these typos throughout the text.
*All those points have been modified in the text.*

- line 266: "3D" => "three-dimensional (3D)"
*This is changed.*

- line 267: What is the RegCM model? I don't see any description of the model.
*This regional model is effectively not used in the study but serves for some comparisons. We have now included in the text the recent reference of Giorgi et al. (2023) that describes this regional climate model. The sentence has been changed as follows : « ...CNRM-CM simulations is consistent with the results obtained by Solmon et al. (2021) who used the RegCM model (Giorgi et al., 2023), but smaller in magnitude.»*

*Giorgi, F., Coppola, E., Giuliani, G., Ciarlo, J. M., Pichelli, E., Nogherotto, R., et al., The fifth generation regional climate modeling system, RegCM5: Description and illustrative examples at parameterized convection and convection-permitting resolutions. Journal of Geophysical Research, 128(6). https://doi.org/10.1029/2022JD038199, 2023.*

- line 297: Define "GG"
*This refers to Gulf of Guinea. This is now clearly indicated.*

- ine 298: Define "O-A"
*This refers to Ocean-Atmosphere. This is now indicated.*

- line 344: "layer). for" => "layer) for" (Remove ".")
*This is changed.*

- line 409: "kg.kg.s^-1" => "kg kg^-1 s^-1"
*This is modified.*

General comments:

This is a second review of 'Impact of Biomass Burning Aerosols (BBA) on the tropical African climate in an ocean-atmosphere-aerosols coupled climate model' after authors revised in response to the first round of referee comments, and comments largely focus on the new or altered content. The authors' additional comments and especially new simulations provided addition context to help disentangle effects of the dynamic SST and the aerosol radiative effects on cloud properties and ERFs, improving the quality of the analyses against the first version. I recommend acceptance after minor revisions.

**We would like to thank the reviewer for his comments. In the new version, we have taken them into account.**

Specific comments:

Minor:

The simulations to separate the impacts from BBA radiative dynamical effects vs. surface SST effects on cloud properties are very useful. As these are the dominant proposed mechanisms at play here, the differentiated models for BBA and SST effects should be mentioned in more places as appropriate through the work to justify claims of whether one effect or the other is dominant. For example:

- 346: Doesn't figure A6a) show that the BBA effect is strongly tied to the increased LCF over the Gabon/Angola coast? Is the increased moisture advection tied to BBA radiative impacts then?
*This is right and we have now clarified this point and changed the sentence as follows : « As shown in Figure A6, the response of the low-level clouds simulated in the coastal areas of Gabon and Angola is not sensitive to the coupling between the ocean and the atmosphere and is probably more related to the increase in moisture advection over this region due to the BBA radiative effect(Figure A7 and A8) that favors north-westerly anomalies over the ocean south of the Equator (see Part 3.3). This contributes to enhance the low cloud fraction.»*

- 475: The BBA effect being weaker isn't simply 'more likely', it is explicitly backed up by figure A6. I recommend mild reorganization to move the claim and the evidence (~line 487) closer together.
*We have now changed the sentence by removing « more likely » and linked the result to the contribution of SST and BBA radiative effects: «As shown in Figures 9a,b,c and A8, this dimming is largely due to the increase of the low-level CF, LWP and COD (mainly controlled by the persistent SST cooling, see the following paragraph), while the BBA direct effect  plays a smaller role in this season.»*

513: I read this sentence as claiming that the SST decrease is the dominant effect on changing cloud patterns, but figure A6 seems to show the opposite-- the BBA effect is at least as strong in JJA, if not stronger, over a wider ocean area and is the dominant influence near Gabon.
*This is effectively right and the sentence has been changed as follows: «This positive impact is found to be mainly due to BBA radiative effect (especially the lower tropospheric heating) associated to a lesser extent to the SST decrease (which is in response to the surface BBA radiative forcing ~-5 to -15 W.m−2 and the cloud changes), . These both contribute to (i) increasing the LTS and (ii) to limiting the intrusion of dry air at the cloud top.»*

Technical corrections:

23: feedback should be 'effect' on precipitation
*This is done.*

142: missing left parenthesis for Druge 2019
*This is now changed.*

149: Biomass Burning shouldn't be capitalized
*This is changed.*

181: should read "the present configuration allows us to focus on solely the…" with no comma
*This is now changed.*

206: favoring should be 'favor'
*This is modified.*

276: 'strongest' should probably be 'stronger'
*This is changed.*

304: define 'GG' and 'O-A' coupling
*These two terms are now defined.*

314: 'with an impact of up to 15W' as a phrase doesn't make sense. '…with an impact of up to [magnitude] at 15W', or perhaps 'with an impact out to 15W' as possible alterations.
***Thank you for the suggestion. We have now used : « with an impact out to 15W ».***

Section 3.3: should read 'dynamics', not 'dynamic'
***This is modified.***

Figure A8 shouldn't have underscores in the title
***This is now changed in the Figure.***

400: Clarify the direction of the model bias with '…biased towards underestimating low-level cloud cover', as long as that is what's intended
***This is changed.***

420: figure reference should be A7
***Thank you, this is changed.***

461: should read 'do not allow us to disentangle…'
***This is modified.***

474: Figure reference should probably be A9, not A8
***Thank you, this is changed.***

487: remove 'the' before 'Figure A6'.
***This is done.***

489: I believe the reference to figure A7 should probably be A6 for SON cloud anomalies
***This is right and now modified.***

498: Clarify to say there are no major changes over the African continent. South America shows significant differences in precipitation.
***These two points are now indicated.***

559: either write 'using different GCMs' or, less favored because it's redundant with the acronym (global circulation model models), 'using different GCM models'
***This is modified.***

508: indicate should be 'indicates'
***This is changed.***

513: Suggest a rephrase to clarify and organize, the original sentence is confusing about what is cause and what is effect(s). One suggestion: This positive impact is found to be mainly due to the SST decrease, which is in response to the surface BBA radiative forcing ($\sim$-5 to -15 W.m$^{-2}$) and the cloud changes associated with lower tropospheric heating. These both contribute to (i) increasing the LTS and (ii) to limiting the intrusion of dry air at the cloud top.
***Thank you for the suggestion. This is now changed.***